# A chemogenetic approach for dopamine imaging with tunable sensitivity

Marie A. Labouesse [1,2,3,11], Maria Wilhelm[1,10,11], Zacharoula Kagiampaki [1,11], Andrew G. Yee [4,11], Raphaelle Denis [5,6], Masaya Harada [1], Andrea Gresch[1], Alina-Măriuca Marinescu[3], Kanako Otomo[3], Sebastiano Curreli [7], Laia Serratosa Capdevila [1], Xuehan Zhou [1], Reto B. Cola [1], Luca Ravotto [1], Chaim Glück [1], Stanislav Cherepanov [8], Bruno Weber [1,2], Xin Zhou[9], Jason Katner[9], Kjell A. Svensson[9], Tommaso Fellin [7], Louis-Eric Trudeau [5,6], Christopher P. Ford[4], Yaroslav Sych[8] & Tommaso Patriarchi [1,2] ✉

Genetically-encoded dopamine (DA) sensors enable high-resolution imaging of DA release, but their ability to detect a wide range of extracellular DA levels, especially tonic versus phasic DA release, is limited by their intrinsic affinity. Here we show that a human-selective dopamine receptor positive allosteric modulator (PAM) can be used to boost sensor affinity on-demand. The PAM enhances DA detection sensitivity across experimental preparations (in vitro, ex vivo and in vivo) via one-photon or two-photon imaging. In vivo photometry-based detection of optogenetically-evoked DA release revealed that DETQ administration produces a stable 31 minutes window of potentiation without effects on animal behavior. The use of the PAM revealed region-specific and metabolic state-dependent differences in tonic DA levels and enhanced single-trial detection of behavior-evoked phasic DA release in cortex and striatum. Our chemogenetic strategy can potently and flexibly tune DA imaging sensitivity and reveal multi-modal (tonic/phasic) DA signaling across preparations and imaging approaches.

Dopamine (DA) is an important neuromodulator known to play pivotal roles in the regulation of animal behavior[1,2]. Loss or dysregulation of DA signaling on target circuits has been linked to several brain disorders, including Parkinson's disease, schizophrenia, and addiction[3]. The use of experimental techniques such as microdialysis and electrochemistry has provided useful insights on the mechanisms of DA release and reuptake in vivo[4], but suffers from limitations, as these approaches can only provide single-point measurements and cannot differentiate signals with cellular or subcellular resolution[5]. In contrast, the recent development of genetically encoded DA sensors (hereafter DA sensors) made it possible to monitor DA dynamics with high spatiotemporal resolution through a variety of optical techniques[6,7].

These sensors inherit the intrinsic pharmacological properties of the parent G protein-coupled receptors (GPCRs) that they are based on, except that they lose the ability to couple with downstream

[1]Institute of Pharmacology and Toxicology, University of Zürich, Zürich, Switzerland. [2]Neuroscience Center Zurich, University and ETH Zürich, Zürich, Switzerland. [3]Department of Health Sciences and Technology, ETH Zürich, Zürich, Switzerland. [4]Department of Pharmacology, University of Colorado School of Medicine, Aurora, CO, USA. [5]Department of Pharmacology & Physiology, Faculty of Medicine, SNC and CIRCA Research groups, Université de Montréal, Montréal, QC, Canada. [6]Department of Neurosciences, Faculty of Medicine, SNC and CIRCA Research groups, Université de Montréal, Montréal, QC, Canada. [7]Optical Approaches to Brain Function Laboratory, Istituto Italiano di Tecnologia, Genova, Italy. [8]Institute of Cellular and Integrative Neuroscience, University of Strasbourg, Strasbourg, France. [9]Eli Lilly and Company, Indianapolis, IN, USA. [10]Present address: Institute for Neuroscience, ETH Zurich, Zurich, Switzerland. [11]These authors contributed equally: Marie A. Labouesse, Maria Wilhelm, Zacharoula Kagiampaki, Andrew G. Yee. ✉e-mail: patriarchi@pharma.uzh.ch

signaling partners[8–10]. Accordingly, their ligand affinity cannot be enhanced through pre-coupling with G proteins, as in the case of some GPCRs[11], and remains in a fixed low-affinity state. Across brain areas, but also within individual areas, physiological DA levels can vary quite extensively on a moment-to-moment basis, depending on the density of DArgic innervation[12], the firing activity of DA neurons (tonic versus phasic firing modes[13,14]), as well as other physiological factors, such as local cholinergic modulation of DA release on DA terminals[15,16]. However, due to their pre-determined affinity and sensitivity range, existing DA sensors fail to capture well both tonic and phasic extracellular DA levels in a single recording session. Furthermore, each indicator performs optimally within a limited range of extracellular [DA] that is dictated by the indicator's fixed affinity and sensitivity. Thus, an approach that would enable capturing the complexity of endogenous multi-modal DA release and, at the same time flexibly, enhance the indicator's detection sensitivity would be ideal to complement current DA sensor-based approaches used to investigate DA physiology.

Positive allosteric modulators (PAMs) act by tuning the affinity of a ligand for its receptor without altering maximal ligand efficacy. These agents are attracting increasing attention in drug development, due to multiple reasons, such as their lack of effect on the target receptor in the absence of its endogenous ligand, their higher pharmacological specificity due to the low conservation of allosteric binding pockets, and the consequently lower risk for adverse consequences upon overshooting a maximal dose[17,18]. In principle, these drugs could be employed in combination with GPCR-based sensors to boost their ligand sensitivity; however, such an approach has not yet been explored as a means to potentiate the sensitivity of sensor-based imaging applications.

Here we demonstrate that a PAM selective for the human DA type-1 receptor (hmDRD1), hereafter referred to as D1-PAM, can boost the affinity of a hmDRD1-based genetically encoded sensor (dLight1.3b) without affecting mouse DRD1 (msDRD1)-dependent neuronal activity and behavior owing to its inherent ultra-low affinity for the msDRD1. We show that this selective chemogenetic strategy can be exploited for tuning the sensitivity of DA imaging on-demand. We characterize the in vitro, ex vivo, and in vivo effects of this strategy for DA sensor potentiation by determining the magnitude, duration, and specificity of the effect and the lack of interference with mouse physiology. Our experiments demonstrate the utility of this approach for boosting DA imaging sensitivity on-demand with multiple imaging modalities, enabling the simultaneous optical detection of tonic and phasic DA release across brain areas in living animals.

## Results

### Conceptualization and in vitro characterization of the approach

To test the hypothesis that the DA affinity of a hmDRD1-based genetically-encoded DA sensor could be boosted in the presence of a D1-PAM (Fig. 1a), we performed DA titrations on HEK293 cells expressing dLight1.3b, the dLight sensor variant with largest dynamic range but lowest DA sensitivity ($EC_{50}$)[9], in the presence or absence of a panel of previously described D1-PAMs[17,19]. The application of D1-PAM compounds MLS1082 and MLS6585 (5 μM) did not have a noticeable effect on the DA sensitivity of dLight1.3b, while the application of DETQ [2-(2,6-dichlorophenyl)−1-((1S,3R)−3-(hydroxymethyl)−5-(2-hydroxypropan-2-yl)−1-methyl-3,4-dihydroisoquinolin-2(1H)-yl)ethan-1-one] produced a strong positive allosteric effect on the sensor (Supplementary Fig. 1a). Analysis of DA titrations performed in the presence of increasing DETQ concentrations (1 nM to 450 nM) based on allosteric $EC_{50}$ shift revealed that the D1-PAM had an allosteric affinity constant (Kb) of 54 nM and induced a maximal ~8-fold leftward shift (α factor = 8.6) in the $EC_{50}$ of DA for the dLight1.3b sensor, decreasing it from roughly 2 μM to 244 nM (Fig. 1b). Given our in vitro results, the known drug-like properties of DETQ (i.e. good blood-brain barrier penetration, in vitro and in vivo efficacy)[20], the fact that it has a known allosteric binding site on its target receptor[21–23], and displays

potent allosteric activity only on the human DRD1 (hmDRD1, $Kb_{human} = 11.4$ nM [DETQ]) while it is >30-fold less potent on msDRD1 ($Kb_{mouse} = 312$ nM [DETQ])[17,21,24], we reasoned that this particular D1-PAM would be an ideal compound for selective chemogenetic potentiation of a human receptor-based DA sensor in a mouse preparation (Fig. 1a).

Two recent studies solved the active state structure of the DRD1 in complex with DA (DA) and LY3154207[22,23], a structural congener to DETQ that is currently in phase-2 clinical trial, clearly identifying the receptor intracellular loop-2 (ICL2) as the site of action for this class of D1-PAMs. Previous work also characterized some of the residues involved in the DRD1-DETQ interaction and indicated that the L143M mutation could boost DETQ potency based on a cAMP readout[21]. We screened a panel of selected mutations targeting the DETQ binding pocket of dLight1.3b in an effort to identify a sensor variant with increased DETQ potency (Supplementary Fig. 1b). While most of the mutations we introduced were detrimental to the allosteric effect of DETQ, we identified one variant (L143I) that showed 3-fold increased DETQ potency (Kb = 17.7 nM; Supplementary Fig. 1c, d). Upon potentiation by DETQ, dLight1.3b$_{L143I}$ reached an $EC_{50}$ for DA of 142 nM in comparison to 244 nM for the non-mutated sensor. Therefore, this sensor variant requires a lower concentration of DETQ to reach a similar degree of positive allosteric effect (α factor = 10.8) and a higher affinity for DA in the potentiated state compared to dLight1.3b. As part of our screening efforts, we also identified a mutated sensor carrying the two mutations L143A and R130Q. This sensor, which we named AlloLite-ctr, was insensitive to DETQ (α factor = 1.6; Fig. 1c, d, f) and thus could be used as a control sensor for subsequent in vivo validation efforts.

Previous work established the target selectivity of DETQ for hmDRD1 against a series of over 40 human protein targets[24,25]. To further confirm its selectivity, we tested the effect of DETQ either in allosteric modulator mode or in agonist mode on mini-G protein recruitment for a subset of mouse receptors using a Nanoluciferase (NanoBit)-based complementation assay[26,27]. Addition of DETQ (100 nM) during DA titrations performed on hmDRD1 or msDRD1 led to a strong and significant potentiation of mini-Gs recruitment only in the case of the hmDRD1, as expected ($P = 0.0156$, Two-tailed Student t-test with Welch's correction; Supplementary Fig. 2a). No allosteric effect was detected also by applying a high concentration of DETQ (1 μM) on all other mouse receptors tested: DRD2, DRD5, ADRA1a, β2AR and 5HT6R (Supplementary Fig. 2b, c). Furthermore, no direct agonist effect was detected when applying DETQ (1 μM) alone on any of the receptors tested (Supplementary Fig. 2d). Thus, our results support previous findings indicating that DETQ acts a PAM selective for hmDRD1.

An important feature of PAMs is that they do not alter the maximal efficacy of the orthosteric ligands, and ideally, this feature should be preserved in our approach as it would greatly ease its implementation and the interpretation of its results. We found that the maximal fluorescent responses to a saturating concentration of DA (100 μM) of dLight1.3b, dLight1.3b$_{L143I}$, and AlloLite-ctr were not affected by a wide range of DETQ concentrations (Supplementary Fig. 3a). Furthermore, direct application of DETQ in the absence of DA could not trigger a response from the sensors (Supplementary Fig. 3b) indicating that DETQ per se does not activate the sensor. Application of DETQ to dLight1.3b in the presence of a sub-maximal concentration of DA (100 nM) rapidly triggered a strong and significant increase in sensor fluorescence (Supplementary Fig. 3c), reflecting the increased sensitivity of the indicator. We next evaluated the DA-selectivity of dLight1.3b in the presence of DETQ. Interestingly, along with potentiation of sensor responses to DA, we also observed potentiation of its responses to noradrenaline (NE), although in the potentiated state the sensor maintained roughly the same degree of DA/NE selectivity (87-fold), as compared to its non-potentiated state[9] (Supplementary Fig. 3d). Furthermore, similar titrations performed on dLight1.3b$_{L143I}$

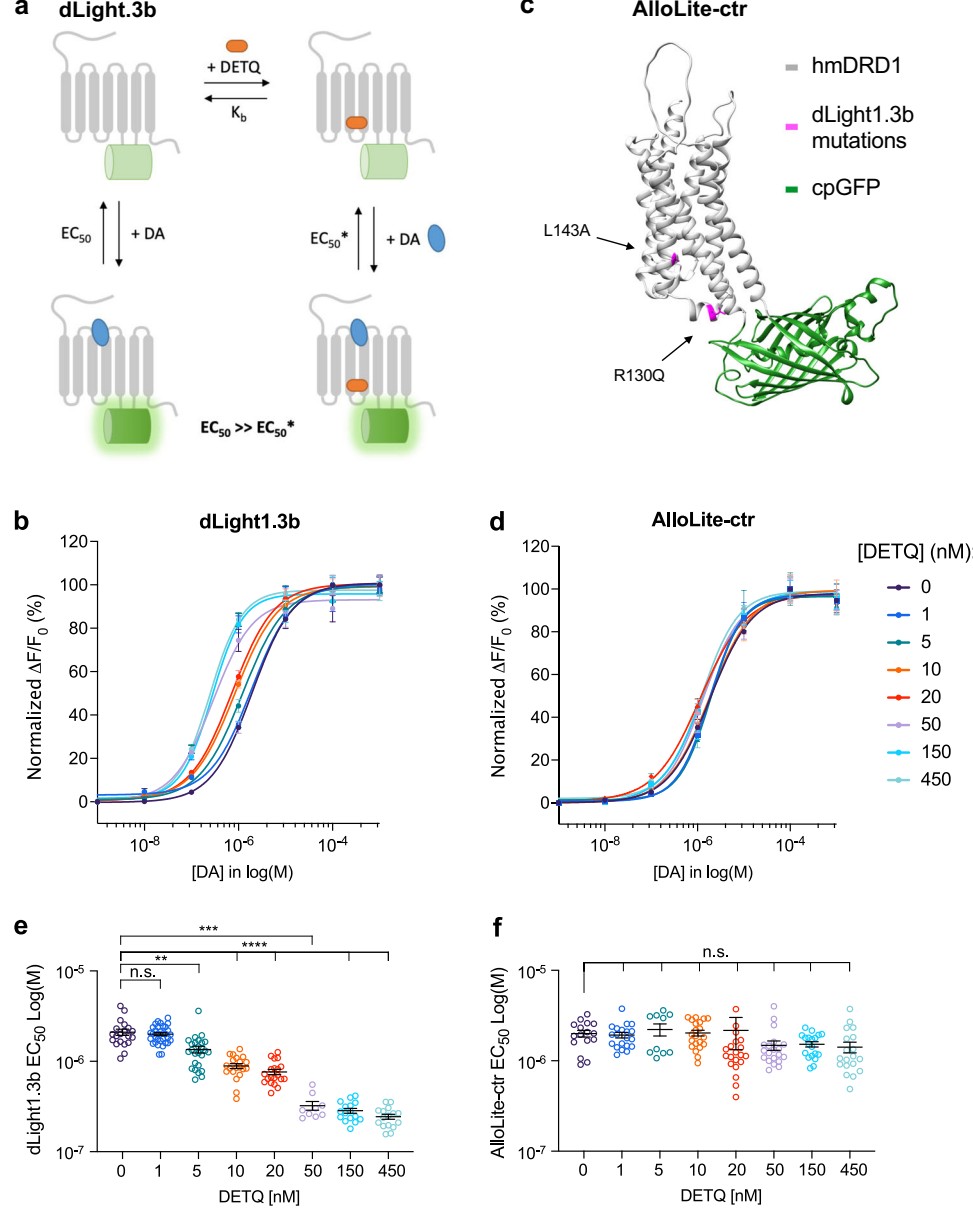

**Fig. 1 | Characterization of the approach and engineering of AlloLite-ctr.**
**a** Schematic illustration of the potentiation strategy. DETQ is the D1-PAM used in this approach, which is selective for human DRD1. Kb, affinity constant of the allosteric ligand. EC$_{50}$, apparent affinity of the orthosteric ligand. EC$_{50}$*, apparent affinity of the orthosteric ligand in the DETQ-bound state. **b**, Dose-response curves obtained from DA titrations on dLight1.3b-expressing cells in the presence of increasing concentrations of DETQ. Data are shown as normalized fluorescence fold-change over baseline ($\Delta F/F_0$ %). Datapoints were fitted using a log(agonist) vs. response nonlinear fit (four parameters) to determine EC$_{50}$ values and using an allosteric EC$_{50}$ shift for determining the alpha factor and the Kb. $n$ = 20, 33, 26, 20, 20, 9, 16, 16 cells for 0, 1, 5, 10, 20, 50, 150, 450 nM DETQ, respectively. **c**, Structural model of AlloLite-ctr, generated using Alphafold2[88]. Residues mutated from

dLight1.3b are highlighted in magenta. **d** Same as in **b** for AlloLite-ctr. $n$ = 16, 20, 10, 21, 21, 20, 19, 19 cells for 0, 1, 5, 10, 20, 50, 150, 450 nM DETQ, respectively. **e**, Plot showing dLight1.3b EC$_{50}$ values measured from **b**. n.s., not significant. $p$ = 0.641; **$p$ = 0.0044; ****$p$ = 2.31 × 10$^{-6}$; ****$p$ = 2.31 × 10$^{-6}$; ***$p$ = 3.01 × 10$^{-4}$; ****$p$ = 1.54 × 10$^{-8}$;****$p$ = 7.8 × 10$^{-8}$ for 0 vs. 1, 5, 10, 20, 50, 150, 450 nM DETQ, respectively. **f**, Same as **e** for AlloLite-ctr (**d**). $p$ = 0.971; $p$ = 0957; $p$ = 0985; $p$ = 0.985; $p$ = 0.3742; $p$ = 0.0711; $p$ = 0.2096 for 0 vs. 1, 5, 10, 20, 50, 150, 450 nM DETQ, respectively. Both **e** and **f** were analysed using parametric one-way ANOVA with Holm-Šídák's multiple comparisons test. All data are shown as mean ± SEM. All experiments were repeated at least 3 independent times with similar results. See also Supplementary Figs. 1 and 2. Source data are provided as a Source Data file.

revealed that this sensor had 53-fold selectivity and a reduced maximal response to NE as compared to DA in its potentiated state (Supplementary Fig. 3d). Finally, application of a number of other neurotransmitters on dLight1.3b-expressing cells in the presence of DETQ did not cause any response from the indicator (Supplementary Fig. 3e), indicating that the molecular specificity of the indicator remained unaffected by the compound. Overall, these results demonstrate that DETQ could be used as a viable tool to potentiate the sensitivity of

dLight1.3b without compromising its dynamic range or ligand-selectivity, and that screening efforts could be deployed in the ICL2 region to further increase sensor affinity for DETQ.

## Imaging DA release from cultured DA neurons with tunable sensitivity
Two recent studies demonstrated the optical detection of spontaneous DA release from cultured DArgic neurons with high spatial

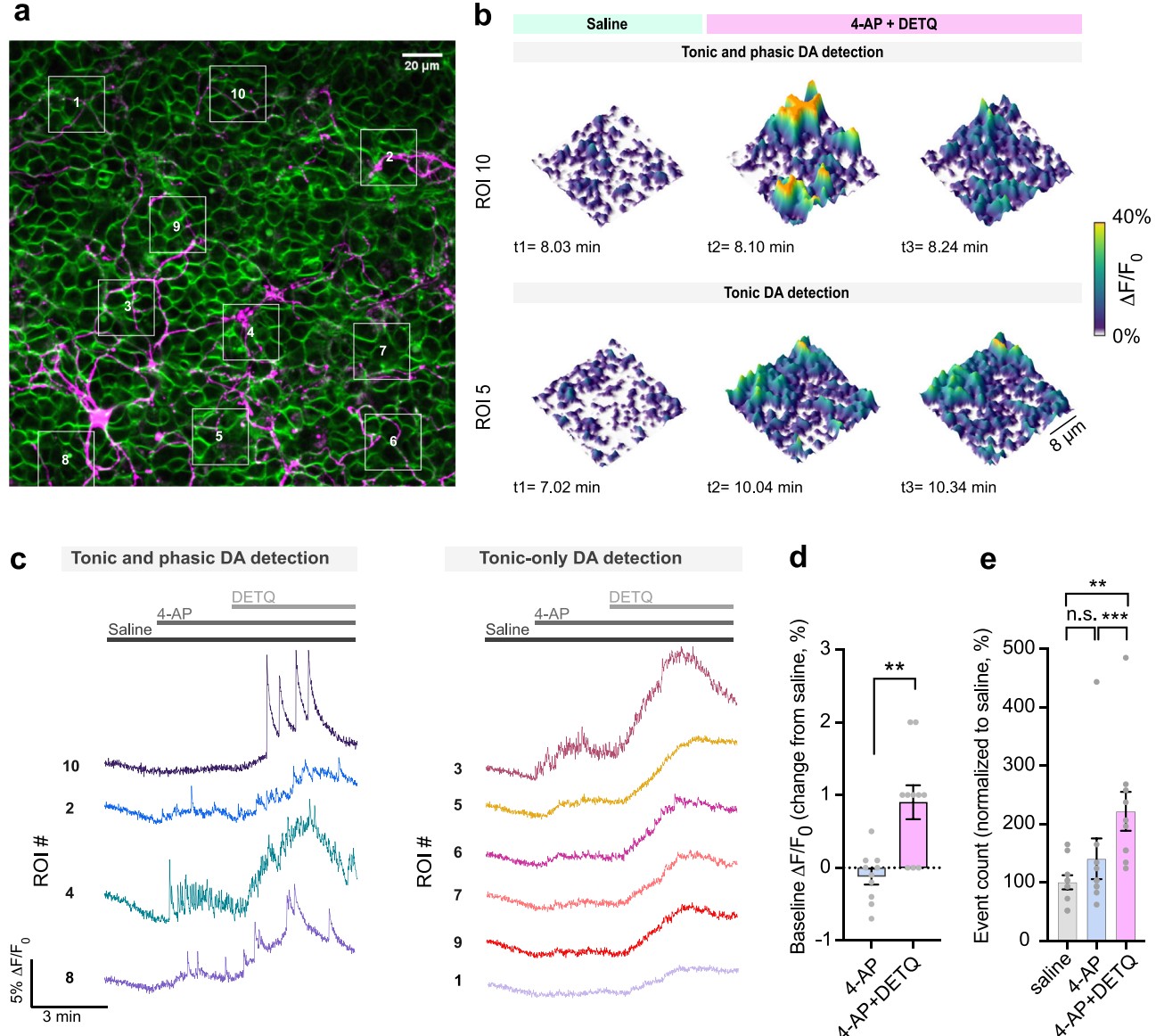

**Fig. 2 | High-resolution imaging of DA release in vitro with tunable sensitivity.**
**a** Representative image of dLight1.3b-expressing HEK293T cells ("sniffer cells",
green) cultured onto mouse primary DArgic tdTomato-expressing neurons
(magenta). White boxes indicate regions-of-interest (ROIs) selected for analysis.
Scale bar, 20 μm. **b**, 3D heatmaps representing dLight1.3b fluorescence reponse
($\Delta F/F_0$) at the different timepoints indicated. Top, sensor responses to a phasic DA
release event observed within ROI 10. Bottom, gradual responses showing an
increase in tonic DA levels detected in ROI 5. Color map represents $\Delta F/F_0$. **c**, Left,
representative traces of dLight responses ($\Delta F/F_0$) from selected ROIs shown with
corresponding numbers in **a** reporting tonic and phasic DA fluctuations (left panel).
Right, a subset of ROIs from which tonic-only DA fluctuations can be seen.
**d**, Comparison of tonic $\Delta F/F_0$ signal detection upon application of DETQ. Wilcoxon
two-tailed matched pairs test, $n = 10$ experiments, $P = 0.0020$. **e**, Comparison of
detected release events between baseline, application of 4-AP, and 4-AP + DETQ.
Each data point represents the sum of events detected per experiment per con-
dition. Data in **d** and **e** is shown as mean ± SEM. Multiple comparisons with Bon-
ferroni correction (a = 0.01667 after correction), $n = 10$ independent experiments.
$p = 0.2998$, 0.007289, 0.001697, for 4-AP vs. saline, 4-AP + DETQ vs. saline, and 4-
AP + DETQ vs. 4-AP, respectively. See also Video S1. Source data are provided as a
Source Data file.

resolution using near-infrared fluorescent carbon nanotubes[28,29]. While
useful, these approaches require custom-made optical detection sys-
tems for imaging in the short-wavelength near infrared (SWIR) spectral
region that are not readily available to most laboratories. Notably, in its
potentiated state dLight1.3b reaches a DA affinity similar to what is
reported for carbon nanotube systems (ie. an $EC_{50}$ of approximately
250 nM[28,29]), suggesting that our approach could potentially offer the
possibility to detect DA release in vitro with sensitivities comparable to
these previous approaches, while imaging in the visible spectrum. To
test this hypothesis, we implemented our approach in a recently
established in vitro assay in which HEK293T cells stably expressing
dLight1.3b were plated onto DArgic neurons in culture and used as

optical DA reporters or "DA sniffer cells"[30]. To clearly identify DArgic
neurons in vitro, our cultures were prepared from DAT[IRES]cre[31] x Ai9-
tdTomato[32] reporter mice expressing the red fluorescent protein
tdTomato in all DArgic neurons (Fig. 2a). The cells were kept under
constant perfusion flow to enable the subsequent application of
compounds and the fluorescence of the sniffer cells was imaged using
confocal microscopy. During unstimulated conditions (i.e., perfusion
with saline solution) only occasional DA release events could be
detected by the sensor. However, when the voltage-gated potassium
channel blocker 4-aminopyridine (4-AP, 250 μM) was added to the
perfusion to increase neuronal excitability, larger localized DA release
events started to become noticeable (Supplementary Movie 1).

Subsequent application of DETQ (200 nM) to the perfusion increased the number of detected phasic DA release events and induced a long-lasting increase in the sensor baseline signal (Fig. 2b–e, Supplementary Movie 1), reflecting the detection of extracellular tonic DA levels in the potentiated state. These data demonstrate that dLight1.3b-expressing sniffer cells can be used with conventional in vitro microscopy techniques (e.g. using 488 nm lasers) to detect asynchronous and spatially-heterogeneous hotspots of DA release from cultured DArgic neurons. Through allosteric boosting of dLight1.3b, DETQ increases the sensitivity of this assay, permitting increased detection of spontaneous phasic DA release events and tonic DA levels in the extracellular environment.

## DETQ enhances two-photon DA imaging in acute brain slices with no detectable off-target effects

To demonstrate that DETQ can be safely utilized as a potentiator of dLight sensitivity without affecting endogenous DRD1-mediated signaling, we first verified that it did not directly alter the electrophysiological properties of DRD1-expressing neurons. We did so by performing whole-cell electrophysiological recordings of DRD1$^+$ medium spiny neurons (MSNs) in dorsal striatum (DS) brain slices. To unambiguously identify DRD1-MSNs during patch-clamp recordings, we injected an adeno-associated virus encoding for a Cre-dependent red-shifted opsin (ChrimsonR[33]) into the DS of DRD1:cre mice (Supplementary Fig. 4a). Neuronal identity could then be confirmed by verifying action potential responses to brief pulses of red light (630 nm) illumination in acute brain slices (Supplementary Fig. 4b). As a positive control, we monitored the effect of a DRD1 agonist (SKF38393) on DRD1-MSN cell firing and membrane potential in current clamp mode. Superfusion of SKF38393 (1 μM) onto the brain slice during the recordings led to a significant increase in the firing rate during current-injection of DRD1-MSNs (Supplementary Fig. 4c, d), as expected based on previous work[34]. No noticeable changes to the resting membrane potential or excitability of DRD1-MSNs was detected when recordings were performed upon superfusion of DETQ (200 nM, i.e. 4-fold higher than its Kb = 54 nM, the in vitro half-maximal apparent affinity constant of dLight1.3b for DETQ; Fig. 1e) compared to the recordings performed in the absence of the drugs (Supplementary Fig. 4c–f). We next tested whether DETQ affected the DA sensitivity of mouse DRD1-MSNs. Bath application of DA (10 μM) increased the number of action potentials evoked by the current injection compared to control conditions (Supplementary Fig. 4g, h). Super-fusion of DA in the presence of DETQ also increased the number of action potentials fired by DRD1-MSNs, but no significant difference existed between DA with or without DETQ (Supplementary Fig. 4i). Taken together, these results indicate that DETQ has no effect on endogenous DRD1-mediated changes in the excitability of DRD1-MSNs.

To test whether DETQ affects DA uptake, we measured DA release in striatal slices using fast-scan cyclic voltammetry (FSCV) (Supplementary Fig. 5a). Bath application of DETQ (200 nM) had no effect on peak DA release or decay constant (tau), indicating that the D1-PAM had no effect on the mouse DA transporter (DAT) (Supplementary Fig. 5b–d). In contrast, cocaine (5 μM) significantly increased both peak DA release and slowed the decay of DA signals by inhibiting its reuptake (Supplementary Fig. 5d). Thus, DETQ can be safely utilized as a potentiator of dLight sensitivity without affecting endogenous DRD1-mediated signaling or DA release.

As a next step, we investigated the ability of DETQ to potentiate dLight1.3b sensitivity in acute brain slices using two-photon microscopy. To achieve sensor expression in brain tissue, we injected wild-type mice with a dLight1.3b-encoding adeno-associated virus (AAV9.hSynapsin1.dLight1.3b) 3–4 weeks before slice preparation and imaging. We first imaged sensor responses while exogenously applying DA to striatal slices by iontophoresis (Fig. 3a–d). Under control conditions, the smallest application of DA (ejection current: 20 nA, 5 ms)

did not evoke any measurable change in dLight fluorescence (Fig. 3a, b). However, in the presence of DETQ (200 nM), this same application of DA produced a robust increase in $\Delta F/F_0$. The area of dLight1.3b $\Delta F/F_0$ exceeding two standard deviations of noise in preceding frames (>2σ noise) was significantly increased by DETQ (Fig. 3c). Analysis of peak $\Delta F/F_0$ over a range of ejection currents demonstrated a 2.6-fold increase of dLight sensitivity evoked by DETQ (EC$_{50}$ control: 67.8 nA, EC$_{50}$ DETQ: 25.8 nA; Fig. 3d). Additionally, DETQ slowed the decay time of fluorescence changes measured by spot photometry (ejection current: 50 nA; Fig. 3e), as expected given the lower EC$_{50}$. Next, we examined the effect of DETQ on synaptically evoked DA release. Endogenous DA release was initially evoked in the DS by single electrical stimulation (10 μA, 0.5 ms; Fig. 3f–g). Fluorescence changes of dLight1.3b produced by electrically stimulated DA release in striatal slices were strongly enhanced by DETQ (Fig. 3f–i). Both peak $\Delta F/F_0$ and area ( > 2σ noise) of dLight signals evoked by this small electrical stimulus were significantly enhanced by DETQ. Importantly, DETQ allowed the detection of significant changes in fluorescence in slices of the medial prefrontal cortex (PFC) (Fig. 3j–m), a region in which DA neuron projection density is much lower than in the basal ganglia[12] and thus DA is more difficult to measure. While a train of stimuli (5 × 20 μA, 0.5 ms, 25 Hz) did not evoke measurable changes of fluorescence under control conditions, allosteric tuning of dLight1.3b was able to increase its sensitivity and unmask small increases of $\Delta F/F_0$, corresponding to sparse DA release in this region.

## In vivo characterization of the approach

In order to safely and effectively utilize DETQ for in vivo applications, it is important to know its pharmacokinetic (PK) profile in the blood and in the brain. A previous study established the PK profile of DETQ in rodents, but was limited to administration via oral gavage[20]. We set out to determine the PK profile of DETQ upon systemic delivery via the administration route and dose used in this study (intraperitoneal injection, 10 mg/kg). We collected blood plasma and brain (frontal cortex) tissue samples at regular intervals from DETQ-injected animals and quantified via HPLC/MS the free (not albumin-bound) DETQ concentrations (Supplementary Fig. 6a). The results show that already 15 minutes after dosing the animals with DETQ, the drug reached free brain levels of approximately 270 nM and gradually decreased to approximately 110 nM at 45 minutes post-injection. The concentration further decreased below 50 nM at 1-hour post-injection, and the drug was almost completely cleared from the brain at 2 hours post-injection (Supplementary Fig. 6b).

Having established the precise PK profile of DETQ in the mouse brain, we next sought to functionally establish the temporal window of its pharmacological effect on dLight1.3b. To do so we performed further recordings in the PFC. We started by verifying the presence of sparse DA terminals in the PFC area of choice by expressing a cre-dependent ChR2-eYFP in DA neurons (Fig. 4a). To achieve optical control of DA release in the PFC, we then expressed ChrimsonR in DA neurons of the ventral tegmental area (VTA) for optogenetic control of DA release, and performed photometry recordings of dLight1.3b in the PFC (Fig. 4b–d). This preparation allowed us to reliably elicit multiple optogenetically-evoked (opto-evoked) DA release events in vivo and determine the effect of DETQ on the amplitude and kinetics of the signals, as well as to establish the duration of the DETQ effect. Administration of DETQ led to a clear and significant increase both in opto-evoked DA signal amplitude and decay time, which started at 5 minutes and lasted until 45 minutes after DETQ administration (Fig. 4i). To precisely determine the duration of a stable DETQ effect, we investigated the time window in which peak amplitude was steadily elevated at a plateau level. This revealed that DETQ produces a stable increase in DA sensitivity lasting from 5 minutes to 36 minutes (31 minutes in total) after i.p. administration ('DETQ Temporal window' in Fig. 4i, k). Importantly, following the stable potentiated state, DA

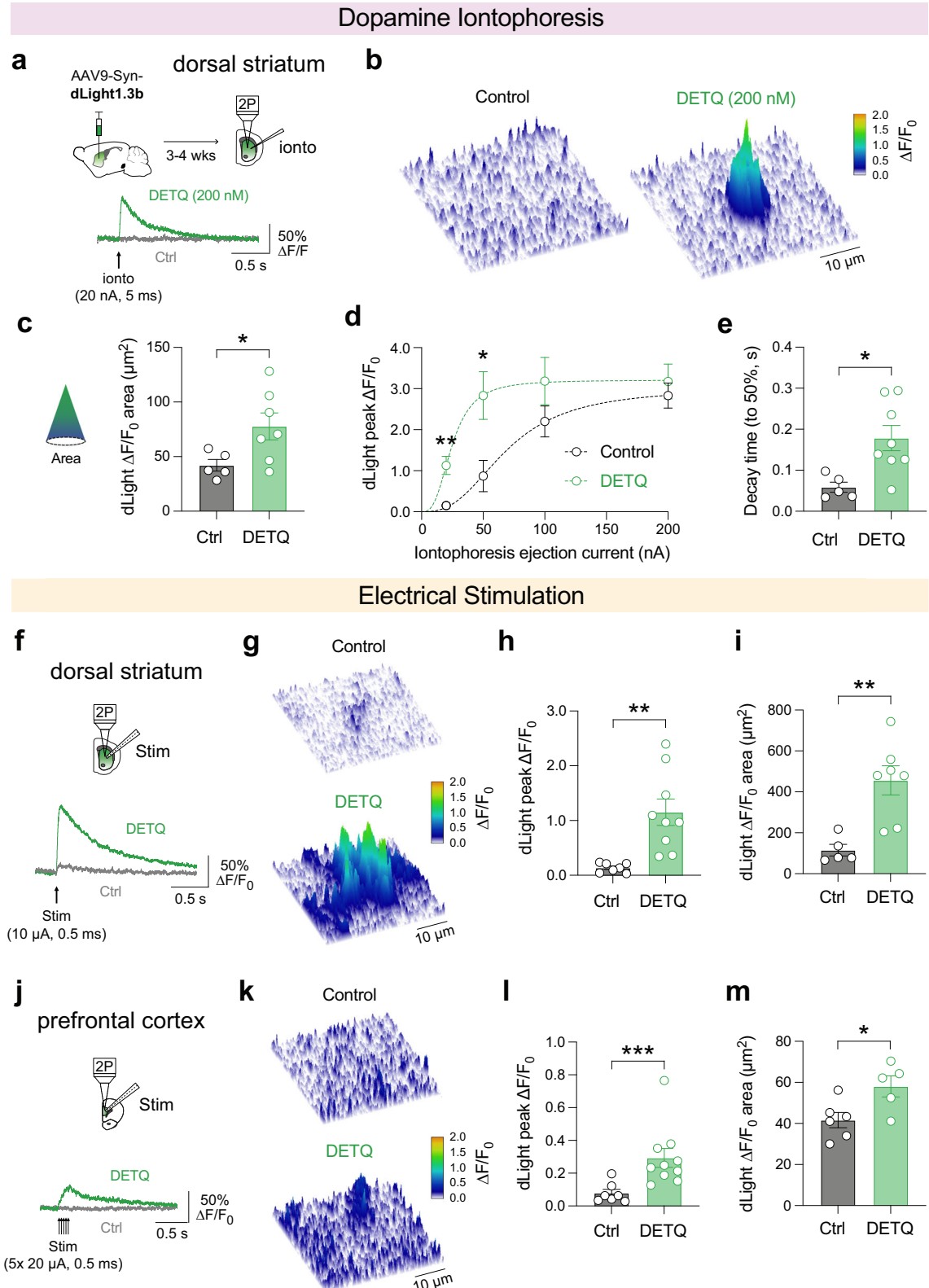

peak amplitudes gradually started to decline and return to a normal non-potentiated state, closely matching the kinetics of DETQ clearance from the brain (Supplementary Fig. 6b).

In order to safely implement DETQ into established behavioral paradigms, it is important that DETQ administration per se does not alter mouse behavior or activity of DRD1 neurons. While previous in vivo work consistently confirmed this in multiple assays (microdialysis-based measurements of neurotransmitter levels; locomotion, cognition, and social interaction tests)[20,24,35,36] and the clearly-established pharmacological mechanism of action and selectivity of DETQ minimizes this possibility[17,19], we performed additional control experiments to ascertain this in our hands. We monitored and quantified animal behavior in an open field arena with a focus on behavioral metrics that can be influenced by DArgic signaling, such as locomotion

**Fig. 3 | Ex vivo two-photon characterization of the approach. a** Experimental schematic and example time-courses of dLight1.3b responses evoked by DA iontophoresis and potentiating effect of DETQ. **b**, Example frames showing the effect of DETQ on peak dLight $\Delta F/F_0$ evoked by DA iontophoresis (ejection current: 20 nA). **c**, Spatial analysis of $\Delta F/F_0$ area exceeding the threshold of >2 standard deviations (2σ) of baseline noise (ejection current: 20 nA). Control: $n = 5$ slices from 3 animals, DETQ: $n = 7$ slices from 3 animals, *$p = 0.0436$, t(10) = 2.31, independent samples t-test (two-sided). **d** Peak dLight $\Delta F/F_0$ evoked by DA iontophoresis with increasing ejection current amplitude. Control: $n = 5$ slices from 3 animals, DETQ: $n = 8$ slices from 3 animals, **$p = 0.0015$, t(7.32) = 4.92, *$p = 0.0135$, t(10.89) = 2.94, Two-way ANOVA followed by Fisher's LSD test. **e**, Decay time (peak to 50%) of dLight signals following DA iontophoresis (ejection current: 50 nA). Control: $n = 5$ slices from 3 animals, DETQ: $n = 7$ slices from 3 animals, *$p = 0.0131$, t(11) = 2.96, independent samples t-test (two-sided). **f**, Example responses produced by single electrical stimulation (1 × 10 μA, 0.5 ms) of DA release in dorsal striatum slices. **g**, Example images of peak responses in the striatum. **h**, Peak dLight $\Delta F/F_0$ evoked by electrical stimulation in striatum. Control: $n = 7$ slices from 3 animals, DETQ: $n = 9$ slices from 3 animals, **$p = 0.0026$, t(14) = 3.66, independent samples t-test (two-sided). **i**, Area of dLight $\Delta F/F_0$ greater than 2σ baseline threshold, evoked by DA release in dorsal striatum. Control: $n = 5$ slices from 3 animals, DETQ: $n = 7$ slices from 3 animals, **$p = 0.0033$, t(10) = 3.83, independent samples t-test (two-sided). **j**, Example dLight $\Delta F/F_0$ traces produced by trains of electrical stimuli (5 × 20 μA, 0.5 ms at 25 Hz) in PFC. **k**, Example images at peak of DA release revealed by DETQ in cortical slices. **l**, Peak dLight $\Delta F/F_0$ evoked by electrical stimulation in PFC. Control: $n = 7$ slices from 3 animals, DETQ: $n = 10$ slices from 3 animals, ***$p = 0.0007$, U = 3, Mann-Whitney U-test (two-sided). **m**, Area of dLight $\Delta F/F_0$ greater than 2σ baseline threshold, evoked by DA release in PFC. Control: $n = 6$ slices from 3 animals, DETQ: $n = 5$ slices from 3 animals, *$p = 0.0267$, t(9) = 2.65, independent samples t-test (two-sided). All data shown as mean ± SEM. Source data are provided as a Source Data file.

and time spent in the center. While cocaine, as expected, caused an increase in the animal's locomotion, DETQ did not have a significant effect at the doses tested (3, 10, 30 mg/kg) (Supplementary Fig. 7a–c). Furthermore, we did not observe any effect of DETQ (10 mg/kg) on total animal locomotion or time spent in the center even after tethering the animals to the optic fiber patchcord used for photometry recordings (Supplementary Fig. 7d–g). In addition, we determined if DETQ affects DRD1-MSN event-related activity in vivo. We used an aversive stimuli known to elicit DA release and to increase NAc DRD1-MSN evoked activity[37,38]. We found that footshock-induced activity in NAc DRD1-MSNs remained unaffected by DETQ injection, but as expected was blunted by a DRD1 antagonist (Fig. S8).

The in vivo photometry recordings performed for the characterization of our approach led us to the intriguing observation that in addition to enhanced detection of individual opto-evoked DA peaks (phasic release), this method could allow us to better detect tonic-like baseline changes in DA. Indeed, soon after DETQ administration, a clear increase in the baseline fluorescence of the sensor could be observed (Fig. 4e, f), which was further boosted by subsequent administration of cocaine (20 mg/kg, Supplementary Fig. 9a–f). This is in agreement with previous in vivo photometry work using a similar indicator[39] as well as with our in vitro observation of the ability of dLight1.3b in the potentiated state to detect tonic DA levels (Fig. 2, Supplementary Fig. 3c). To further investigate this, we performed in vivo photometry recordings using our approach in the nucleus accumbens (NAc) of freely-moving mice, where spontaneous DA release events can be detected using dLight1.3b[40]. These recordings revealed that in addition to the expected increase in the amplitude of spontaneous DA events during the time window of the approach, a noticeable increase in sensor baseline (about 15–20% $\Delta F/F_0$) could be detected, which was larger than the one we previously observed in the PFC (about 2–4% $\Delta F/F_0$, Supplementary Fig. 10a–d), potentially reflecting regional differences in tonic DA levels. Importantly, this baseline increase was observed in the absence of any change in locomotion (Supplementary Fig. 7d–g).

To confirm that this effect was not due to an unexpected increase in tonic DA release upon DETQ administration, we performed control experiments leveraging AlloLite-ctr to measure DA release, as it bears nearly identical composition, dynamic range, and DA sensitivity as the parent sensor (EC$_{50}$ 1.9 vs. 2.1 μM for AlloLite-ctr vs. dLight1.3b, respectively), with the only exception that it does not respond to DETQ (Fig. 1). First, we established the sensitivity of AlloLite-ctr to phasic DA release in the NAc by measuring optogenetically-evoked DA peaks upon VTA-DA neuron stimulation (Fig. 5a–c). Similar to what we observed in the PFC for dLight1.3b, AlloLite-ctr could reliably report opto-evoked DA peaks in the NAc as well as their increased decay time upon administration of cocaine (Fig. 5d–g). However, unlike dLight1.3b (Fig. 4j) the opto-evoked DA peaks of AlloLite-ctr retained the same amplitude and kinetics upon administration of DETQ, demonstrating

that the control sensor cannot be potentiated by administration of DETQ in vivo (Fig. 5f–g). Next, we compared the baseline photometry signals between dLight1.3b and AlloLite-ctr in the NAc upon potentiation. While recordings in dLight1.3b-expressing animals consistently showed a large increase in baseline signal after DETQ administration, this effect was completely absent in AlloLite-ctr recordings (Fig. 5h–n, Supplementary Fig. 10e–h), confirming that DETQ does not promote DA release. Thus, the nature of this step-like increase in dLight1.3b baseline signal reflects its ability to detect lower tonic DA concentrations in its potentiated state. We further confirmed this by investigating the DArgic nature of this dLight signal detected with DETQ. First, we chemogenetically inhibited VTA DA neurons during in vivo photometry recordings of dLight1.3b in the NAc by expressing the inhibitory DREADD hM4Di[41] in the VTA and administering the hM4Di agonist compound J60[42] 10 minutes after the injection of DETQ (Fig. 6a, b). Chemogenetic inhibition dose-dependently decreased the step-like increase in dLight1.3b baseline signal observed with DETQ, indicating that this signal is partially dependent on VTA DA firing activity. A signal decrease, albeit milder in amplitude, was also observed in the absence of DETQ, indicating that DETQ amplifies the detection sensitivity of tonic changes in DA (Fig. 6c, d). We further confirmed the DA nature of this signal in an optogenetic setup, where direct optogenetic inhibition of VTA DA axons within the NAc with eNpHR[43] was sufficient to decrease the dLight1.3b fluorescence measured in the presence of DETQ (Fig. 6e–i). Overall these experiments demonstrate that the combination of DETQ and dLight1.3b can be used to achieve on-demand multimodal detection of phasic and tonic DA within the same brain area without affecting endogenous DA release properties.

Furthermore, to provide insights into the stability of these effects, we evaluated the inter-animal as well as within-animal variability in the responses of dLight1.3b to DETQ by repeating recordings once weekly for 3 weeks. Although the step-like increase in baseline after DETQ was variable across animals, within-animal variability across repeated test days was minimal (Supplementary Fig. 11).

## Optical investigation of tonic DA levels across metabolic states and brain regions
To test whether our approach could detect changes in tonic DA levels in vivo in response to physiological challenges, we monitored tonic DA across metabolic states (well-fed versus food-deprived). Rodents are commonly food-deprived to 90% of their baseline body weight (BW) to increase their motivation for task completion in operant behaviors. This metabolic state-dependent increase in motivation is thought to depend on elevated DA levels[44]. For instance, a 12 h fasting period was shown to increase DA release measured with slice voltammetry[44]. Measuring slow, tonic changes in vivo with DA sensors and photometry has been more challenging, given the absence of 'ground truth' baseline in photometry signals. Here we used the 'step-like' increase in dLight1.3b baseline signal following DETQ to measure relative changes

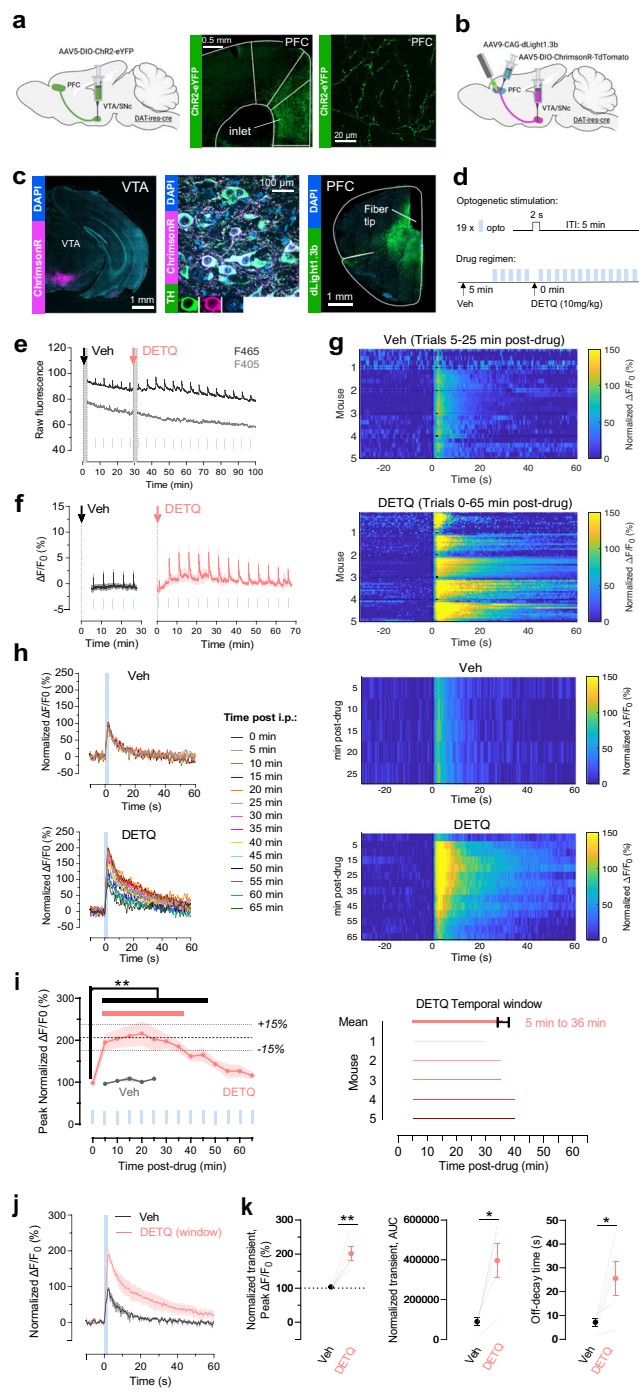

**Fig. 4 | In vivo optogenetic characterization of the approach. a** VTA projections to PFC. **b** Virus strategy for in vivo experiments. Graphics in **a–b** were created with BioRender.com. **c** Representative immunohistochemistry showing VTA ChrimsonR expression and colocalization with tyrosine hydroxylase (TH) and PFC dLight1.3b expression. **d** Experimental timeline of opto-stimulation (2 s, 20 Hz, 20-ms pulses; blue lines) and drug regimen. **e** Representative trace showing raw 465/405 nm-excited fluorescence data after Vehicle/DETQ injection; blue lines: opto-trials. **f** Average traces showing opto-evoked trials (blue lines) over time after Veh/DETQ. **g** Heatmaps aligned to opto-onset showing individual trials ($\Delta F/F0$ normalized to the average Veh trial) at different timepoints post-drug for each mouse in Veh (top) and DETQ (bottom) conditions. **h** Left: average traces (all mice) for opto-evoked trials shown in **g**. Right: Same data as heatmaps. **i**. Left: Average peak normalized $\Delta F/F0$ calculated from Veh and DETQ trials shown in **h**. plotted against time post-drug. Thick black dotted line: value at the plateau of DETQ maximal responses; thin black dotted lines: +/− 15% of the plateau. Thick black continuous line: 'temporal window of superior sensitivity', i.e. 5 to 45 min post-DETQ where peak normalized $\Delta F/F$ after opto-stim is significantly higher than at 0 min post-DETQ (all **$p < 0.01$: 1-way ANOVA: F (17, 68) = 11.56; post-hoc vs. 0 min: 5–35 min $p < 0.0001$, 40 min: $p = 0.0056$, 45 min: $p = 0.0032$). Thick red continuous line: 'temporal window for stable imaging', i.e. 5 to 36 min post-DETQ where DETQ trials have peak $\Delta F/F0$ responses within +/− 15% of the plateau (maximal) response. Right: Individual values of 'temporal window for stable imaging' across mice. Average window: 5 to 36 min post-DETQ (31 min). **j** Average opto-evoked traces in Veh vs. DETQ (trials within 'temporal window for stable imaging'). **k**, Quantification shows significant increase in peak $\Delta F/F0$, AUC and off-decay time in DETQ (temporal window) condition vs. Veh (*$p < 0.05$, **$p < 0.01$). AUC, $n = 5$ mice/group. Two-sided paired t-test, $p = 0.0143$. Peak maxima, $n = 5$ mice/group. Paired t-test, $p = 0.0078$. Off-decay, $n = 5$ mice/group. Paired t-test, $p = 0.0450$. Data shown as mean/SEM. Source data is provided.

in tonic DA concentrations. Mice expressing dLight1.3b in the NAc were injected with vehicle and DETQ 10 min later to detect the step-like increase across 2 different states: ad libitum fed or food-restricted (Supplementary Fig. 12a–c). As seen previously, DETQ led to a consistent and rapid increase of baseline fluorescence. In the food-restriction state, the step-like increase in baseline fluorescence was significantly higher than in the ad libitum state (Supplementary Fig. 12d, e), indicating elevated tonic DA levels, in line with published voltammetry data[44]. This data demonstrates that the use of DETQ can be used to uncover changes in tonic DA signals induced by environmental manipulations.

Next we used multi-fiber photometry[45,46] to test whether our approach would permit the optical detection and comparison of tonic DA levels across different brain regions within the same animal. Mice were implanted with 12 fiber-arrays (3 fibers targeting subregions of

PFC, 4 fibers targeting NAc and 4 fibers targeting caudate putamen CPu/DS). The mice were tethered with a flexible fiber bundle and the fluorescence from dLight1.3b was simultaneously imaged across all channels using a camera sCMOS sensor (Fig. 7a). The recording session consisted of a ~1 hour baseline (before the i.p. injection) and ~2 hours after the injection. On separate days we administered either DETQ or vehicle as a control experiment. DETQ rapidly increased the fluorescence baseline (example recording in CPu showing the increase up to 50% $\Delta F/F_0$ Fig. 7b). Similar to our previous results, upon administration of DETQ, we observed a consistent and rapid increase of baseline fluorescence across all recorded regions expressing dLight1.3b sensor (Fig. 7c). Injection of the vehicle did not alter the fluorescence baseline. We observed different amplitude changes across brain regions (peak $\Delta F/F_0$ fluorescence in NAc was $0.14 \pm 0.16\%$ after the vehicle injection, $22.3 \pm 6.11\%$ after DETQ; in CPu vehicle $−0.7 \pm 0.29\%$, DETQ $28.3 \pm 6.27\%$; and in PFC vehicle $−0.64 \pm 0.39\%$, DETQ $6.8 \pm 2.39\%$; mean ± SEM; ***$p < 0.001$ Mann-Whitney U-test, Bonferroni correction; Fig. 7d) and estimated similar DETQ pharmacokinetic decay times (1/e decay of the peak fluorescence to -0.37 of the peak fluorescence ~30 min – 1 hour after injection; **$0.001 < p < 0.01$, ***$p < 0.001$ Mann-Whitney U-test, Bonferroni correction; Fig. 7e). To identify whether DETQ enhances the $\Delta F/F_0$ variance of the recorded signals we used a wavelet transform to decompose the fluorescence signals into temporal scales. We used a time window of 30 min before the DETQ injection as the baseline recording and 5–35 min after the injection of the compound as DETQ recording. Interestingly, using DETQ we could identify prevalent tonic components specific to the target brain region. We could detect tonic variations of fluorescence signal at temporal scales of 10–20 sec in NAc, 10 sec in CPu and phasic-like component close to 3.5 sec in PFC (Fig. 7f). Dynamic $\Delta F/F_0$ variations at these temporal scales were less obvious, closer to the noise level; however, after administration of DETQ, we could robustly detect such tonic components. Our data demonstrate that DETQ can be used to boost dLight1.3b sensitivity across cortical and subcortical regions and to uncover region-specific temporal dynamics of tonic DA signals.

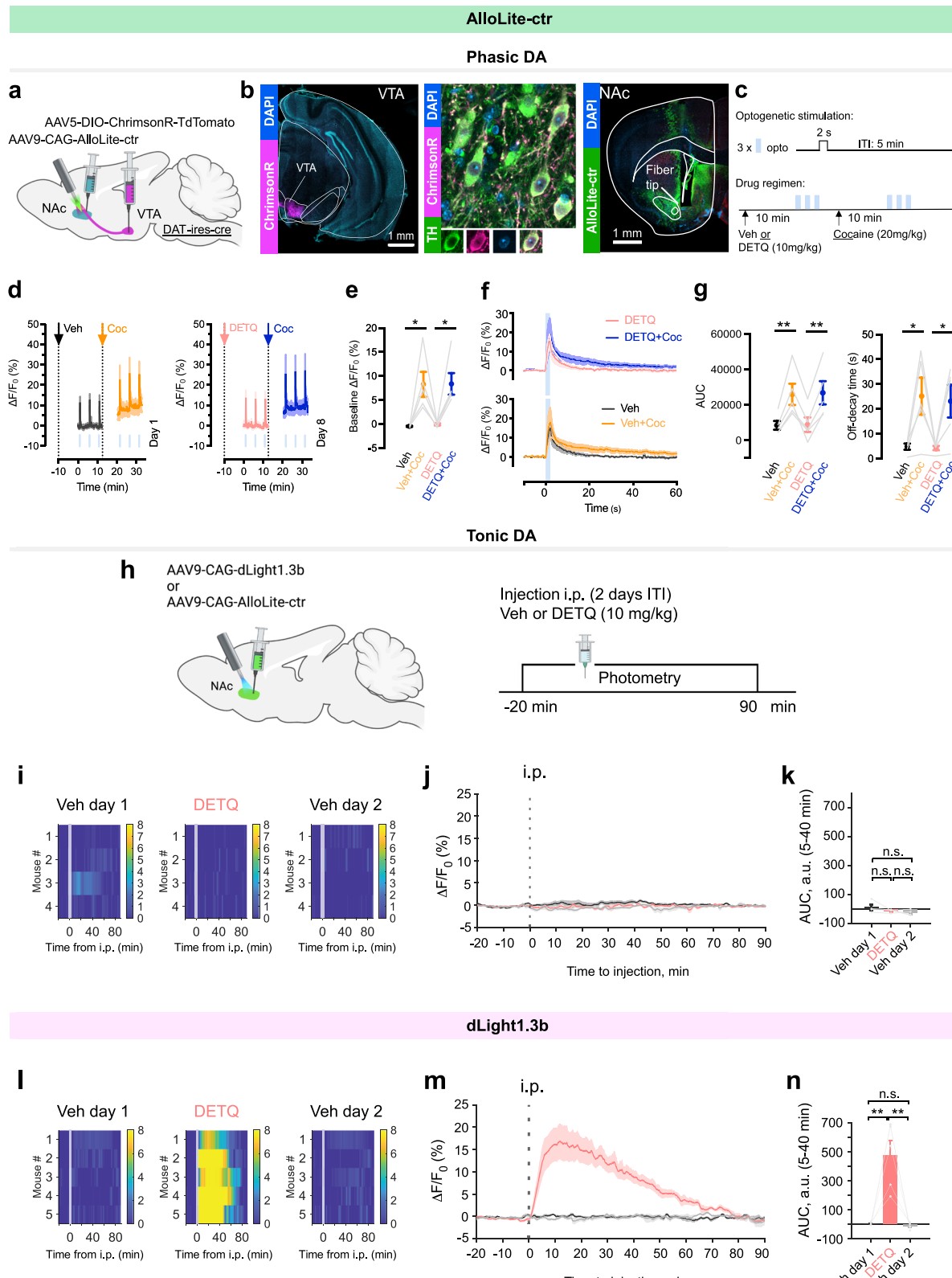

### Enhanced detection of phasic DA release during naturalistic behaviors

To further illustrate the utility of DETQ for enhancing the optical detection of phasic DA release in experimental paradigms involving natural animal behaviors, we tested it in the NAc during a cue-reward Pavlovian conditioning task (Fig. 8a, b). Animals were food-deprived to enhance motivation for reward consumption and were trained in 12 consecutive session days to associate the presentation of an auditory cue (CS, 80 dB, 5 kHz tone, 12 sec) to a lick-initiated liquid reward delivery (US, 30% sweetened condensed milk). Upon completion of training (expert sessions), we performed photometry recordings of dLight1.3b in the NAc during the temporal window for stable imaging defined in Fig. 4i (-10–40 minutes after i.p. injection) upon administration of vehicle or DETQ in interleaved trial days. On DETQ days, With

**Fig. 5 | No effect of DETQ on tonic/phasic DA release properties monitored with AlloLite-ctr in vivo. a–g** Validation of phasic DA detection using AlloLite-ctr. **a**, Viral injection strategy. **b**, Representative immunohistochemistry showing VTA ChrimsonR expression and colocalization with tyrosine hydroxylase (TH) and NAc AlloLite-ctr expression. **c**, Experimental timeline of injections: vehicle (Day 1) or DETQ (10 mg/kg) (Day 8); cocaine (20 mg/kg) 20 min later. Opto-activation of DA axons started 10 min after each injection (3 trials; 2 s, 20 Hz, 20-ms pulses) every 5 min. **d** Average traces showing opto-evoked trials (blue lines) over time after Veh/Cocaine (left) and DETQ/Cocaine (right). **e** Quantification shows significant increase in baseline $\Delta F/F0$ with cocaine vs. Veh and cocaine/DETQ vs. DETQ ($*p < 0.05$). $n = 5$ mice/group, one-way ANOVA, main effect of drug (cocaine): $p = 0.0214$; main effect of drug (DETQ): $p = 0.4011$; drug (cocaine) x drug (DETQ) interaction: $p = 0.8209$. **f** Average opto-evoked traces aligned to opto-onset in all 4 conditions. **g** Quantification of opto-evoked trials shows significant increase in AUC and off-decay time with cocaine vs. Veh and cocaine/DETQ vs. DETQ ($*p < 0.05$;

$**p < 0.01$). AUC: $n = 5$ mice/group, one-way ANOVA, main effect of drug (cocaine): $p = 0.0050$; main effect of drug (DETQ): $p = 0.7645$; drug (cocaine) x drug (DETQ) interaction: $p = 0.7208$. Off-decay: $n = 5$ mice/group, one-way ANOVA, main effect of drug (cocaine): $p = 0.0261$; main effect of drug (DETQ): $p = 0.6262$; drug (cocaine) x drug (DETQ) interaction: $p = 0.7847$. **h** Left, experimental strategy for measuring tonic DA using DETQ. **i** Baseline fluorescence for all AlloLite-ctr mice after Veh/DETQ injections, respectively. **j** Average AlloLite-ctr traces grouped by injection type ($n = 4$ mice). **k** Quantification of baseline $\Delta F/F0$ AUC change for AlloLite-ctr from **i**. $p = 0.49$, Veh-day 1 vs. DETQ; $p = 0.81$, DETQ vs Veh-day 2; $p = 0.23$, Veh-day 1 vs Veh-day 2 ($n = 4$ mice). **l–n**, same as **i–k** for dLight1.3b. $**p = 1.7 \times 10^{-3}$, Veh-day 1 vs. DETQ; $**p = 1.5 \times 10^{-3}$, DETQ vs Veh-day 2; $p = 0.99$, Veh-day 1 vs Veh-day 2 ($n = 5$ mice). One-way ANOVA and Tukey post-hoc tests. See also Figs. S7, S10. Data shown as mean/SEM. Graphics in **a, h** were created with BioRender.com. Source data is provided.

DETQ, we observed a clear and significant increase in the amplitude of both CS and US-associated DA peaks, corresponding to an enhancement factor of approximately 2- and 1.5-fold, respectively, that was visible both at the single-trial level and upon averaging all trials, as compared to vehicle days (Fig. 8c–f). DA peaks were not significantly different between the two Vehicle sessions (Fig. 8d–f). Importantly, both latency to lick and overall licking activity were not affected by DETQ, indicating that the compound does not influence the animal's motivation to collect rewards (Fig. 8d–g).

We next focused on the PFC, where DA release is sparser and harder to detect. Given that DA release in the PFC has been associated with diverse behavioral functions[47], we tested a panel of experimental stimuli carrying either positive or a negative valence (Fig. 8h). We started by presenting the animals with unpredicted rewards separated in time by variable inter-trial intervals (15–75 s) (Fig. 8i). On vehicle days, consumption of the rewards led to a small DA peak that was clearly visible in the all-trials average, but not easily distinguishable on a single-trial basis. Conversely, with DETQ we observed a 2-fold increase in the average peak amplitude vs. Vehicle and we could more easily detect DA peaks over background noise even on a single trial basis, while licking duration remained unchanged (Fig. 8j–n). We then exposed mice to aversive stimuli through delivery of unexpected footshocks. DA signals could be detected also during the presentation of these aversive stimuli, and their amplitude was significantly larger with DETQ, compared to interleaved vehicle days, while the animal's natural response to footshocks (freezing) was not affected by DETQ (Fig. 8n–p). These data demonstrate that DETQ can be effectively deployed during a variety of naturalistic behaviors to enhance phasic DA detection sensitivity up to 2-fold, thus enabling single-trial DA detection even in regions outside the DA-rich limbic system, like the PFC.

## Discussion

In this work we introduce the use of the D1-PAM DETQ to expand the capabilities of an existing genetically encoded DA sensor (dLight1.3b) by enabling on-demand chemogenetic tuning of its affinity. We demonstrate its utility for enhancing DA detection sensitivity in a variety of experimental settings and with different imaging modalities, from cultured DA neurons to acute brain slices and behaving animals. Our in vitro results demonstrate that DETQ can achieve an 8-fold potentiation effect on the $EC_{50}$ of dLight1.3b, while in our in vivo studies we show that systemic administration of DETQ can boost about 2-fold the amplitude of dLight1.3b's fluorescence response evoked by behavioral stimuli. Notably, this approach can only be deployed using hmDRD1-based sensors, given the high species and receptor subtype-selectivity of DETQ. An advantage of our strategy is that it can be directly implemented with an already established and accessible DA sensor that is widely utilized by neuroscience laboratories[40,48–50].

We demonstrate that the high species-selectivity of DETQ renders it pharmacologically inert on mouse physiology, by characterizing its

effects on mini-G protein recruitment in vitro using a panel of mouse receptors, as well as on endogenous DAT activity in mouse brain slices. We also did not observe any effects of DETQ on DA-dependent increases in DRD1-MSN activity both ex vivo and in vivo, on reward-related behavior in a Pavlovian task, on the animals' natural response to unpredicted footshocks (freezing), or on locomotor or anxiety behavior upon in vivo administration in mice. This evidence adds to an extensive literature in which the lack of effects of this drug on mouse neurochemistry, physiology, and behavior had been thoroughly characterized[18,20,21,24,35]. Thus, at the dose we recommend for in vivo dLight1.3b potentiation (10 mg/kg) DETQ can be safely considered pharmacologically-orthogonal to mouse physiology, akin to the concept used in other classic chemogenetic approaches, such as DREADDs[41].

A limitation of our approach is the short duration of the stable effect that DETQ produces in vivo in mice (i.e. 31 minutes), which is determined by the pharmacokinetic properties of DETQ and i.p. administration route used. Because of this, adaptations of the approach may be needed for in vivo experiments in which high-sensitivity DA detection is constantly needed over >30 min or even multiple hours (e.g. during sleep-wake recordings). For example, in the future, it may be possible to overcome the limitations associated with the short duration of the effect and the necessity for i.p. injections through the development and testing of sustained-release formulations[51–53] of the compound perhaps in combination with a different administration route (e.g. *per os*), or alternatively through the use of an osmotic minipump providing stable peripheral administration, or direct microinfusion[54] of the compound in proximity to the recording area. Nevertheless, as we have demonstrated in our results, the DETQ-evoked step-like increase in dLight1.3b signal is overall very consistent and stable when measured within animal and across time. As a consequence, the approach is compatible with a wide range of existing mouse behavioral assays, a sample of which was demonstrated in this study (Pavlovian conditioning, unexpected reward/footshock, open field).

This work has several important implications. First, our demonstration of high-resolution imaging of multimodal DA release from cultured DArgic neurons opens up the possibility of establishing this approach as a screening platform for genetic or pharmacological manipulations that can affect DA release by targeting intrinsic or extrinsic factors in cultured neurons from rodents or potentially even hiPSC-derived neurons. Furthermore, as we demonstrated, the approach can be used to facilitate single-trial detection of DA release in awake-behaving animals even under challenging conditions, such as during DA imaging in the cortex.

Second, the ability to switch on-demand the affinity and sensitivity of dLight1.3b with DETQ makes it possible to simultaneously track tonic and phasic DA release within the same animal and experimental preparation with a 'step-like' readout, which is not possible with current approaches involving the use of genetically encoded sensors,

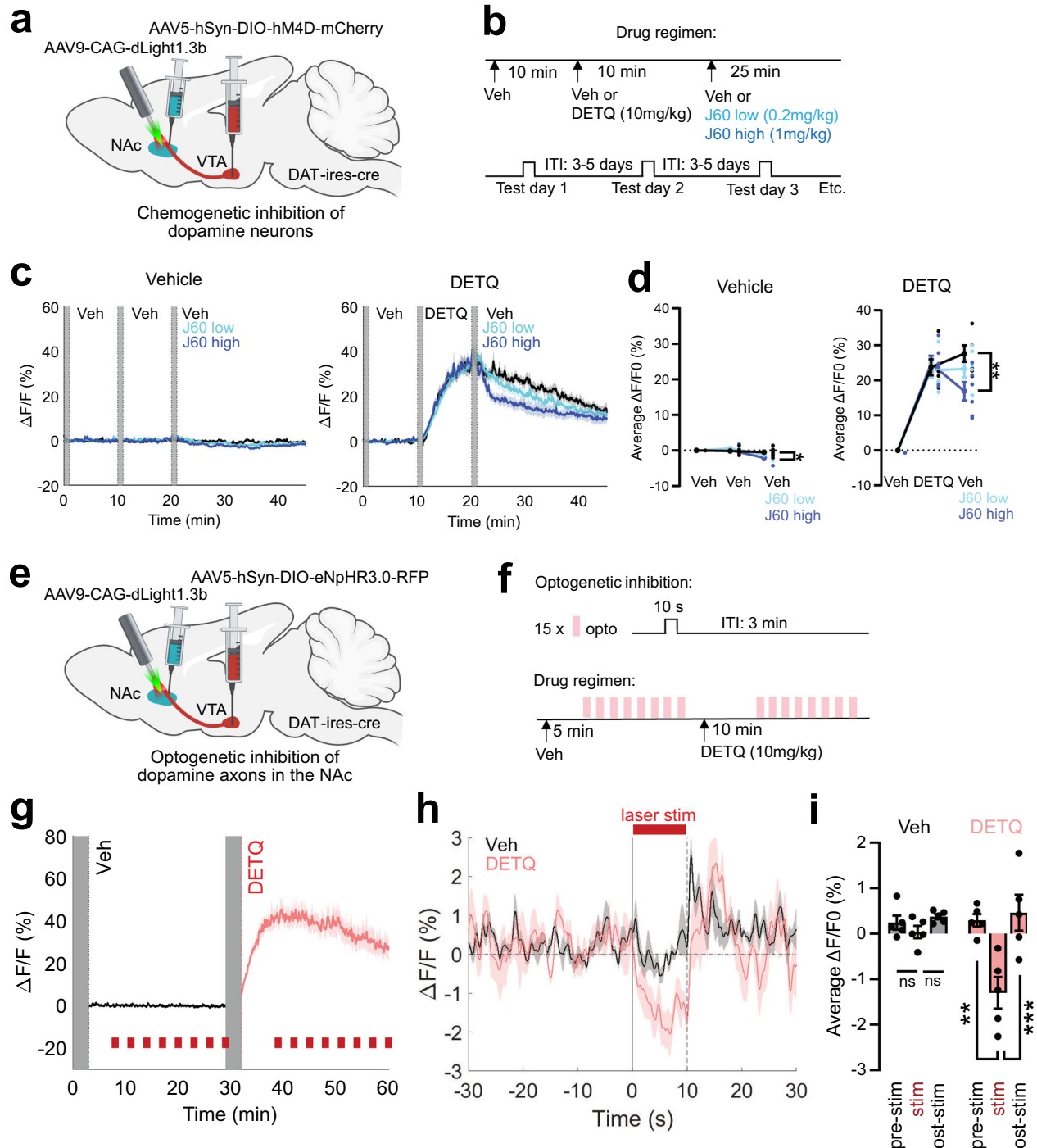

**Fig. 6 | dLight1.3b responses to DETQ in NAc partially depend on the activity of VTA DA neurons. a** Virus strategy for chemogenetic inhibition. **b.** Injections: vehicle (epoch 1); then Veh or DETQ (epoch 2); then Veh or the DREADD agonist J60 (low-dose: 0.2 mg/kg) or J60 (high-dose: 1 mg/kg) (epoch 3). **c.** Average dLight traces with Veh or DETQ as the second drug. DETQ rapidly increases baseline fluorescence. Chemogenetic inhibition of VTA DA neurons decreased dLight signal in the presence of DETQ, confirming its DArgic nature **d.** Quantification of drug-modified dLight signals. Veh: J60-high significantly decreases average dLight $\Delta F/F0$ (*$p < 0.05$). $n = 6$ mice, 2-way RM-ANOVA, drug x epoch: $F(2.154, 10.77) = 5.586$, $p = 0.0202$; Sidak post-hoc on epoch 3: Veh vs. J60-low: $t(5) = 1.786$, $p = 0.2504$; Veh vs. J60-high: $t(5) = 3.876$, $p = 0.0233$. DETQ: J60-high significantly decreases average dLight signal (**$p < 0.01$). $n = 6$ mice, 2-way RM-ANOVA, drug x epoch: $F(1.661, 8.307) = 16.82$, $p = 0.0016$; Sidak post-hoc on epoch 3: Veh vs. J60-low: $t(5) = 1.868$, $p = 0.2269$; Veh vs. J60-high: $t(5) = 5.192$, $p = 0.0070$. **e** Virus strategy for opto-

inhibition of DA NAc axons. **f** Injections: Veh, then DETQ. Opto-inhibition triggered every 3 min (8 trials/drug). **g.** Average dLight traces after Veh or DETQ. DETQ rapidly increases dLight signal amplitude. **h** Average traces showing dLight fluorescence in the −30, + 30 s before/after opto-inhibition. VTA DA opto-inhibition decreases dLight fluorescence, confirming its DArgic nature. Effect only detected in the presence of DETQ, indicating enhanced detection sensitivity for tonic changes in DA levels. **i.** Quantification of opto-modified dLight signals. Veh (left): opto-inhibition ('stim') did not significantly modulate average dLight $\Delta F/F0$. DETQ (right): the same manipulation significantly decreased average dLight $\Delta F/F0$ (**$p < 0.01$, ***$p < 0.001$). $n = 5$ mice, 2-way RM-ANOVA, drug x epoch: $F(2, 8) = 10.93$, $p = 0.0052$; Sidak post-hoc: Veh - 'stim' vs. 'pre': $t(8) = 0.8363$, $p = 0.9647$; 'stim' vs. 'post': $t(8) = 1.356$, $p = 0.7609$. DETQ-'stim' vs. 'pre': $t(8) = 6.483$, $p = 0.0011$; 'stim' vs. 'post': $t(8) = 7.158$, $p = 0.0006$. Data shown as mean/SEM. Graphics in **a,e** were created with BioRender.com. Source data is provided.

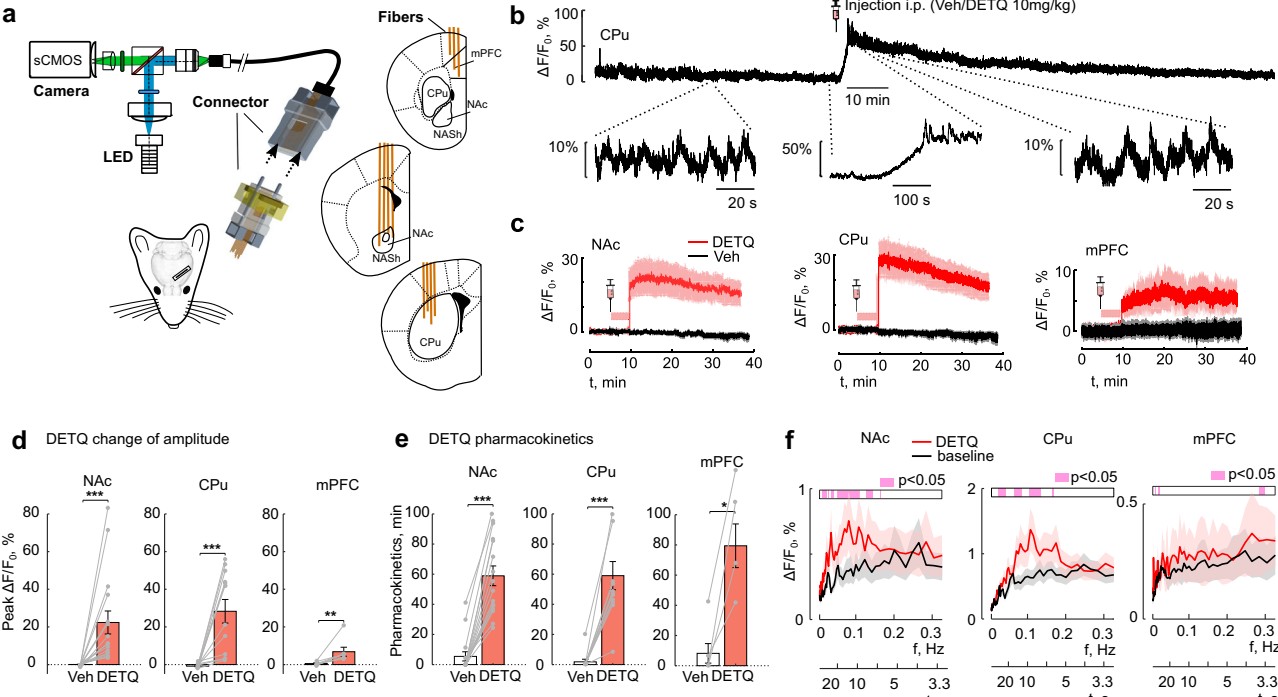

**Fig. 7 | Using DETQ to monitor tonic DA levels across brain areas. a** Schematic of multi-fiber fluorescence recording setup (left) with the implanted 12-fiber array (right). Schematics of coronal brain sections (black) with optical fibers (orange) targeting PFC, NAc and CPu. **b** Example $\Delta F/F_0$ fluorescence traces of dLight1.3b recorded in CPu after DETQ i.p. injection (top, full view). Zoom at the higher temporal resolution on the periods before, during and after the injection. **c** $\Delta F/F_0$ fluorescence of dLight1.3b enhanced by DETQ, pooled across all mice and independent recording sites (mean DETQ red, mean vehicle black; shaded error bar ± SEM; $n = 4$ mice). **d**, Bar plots showing average $\Delta F/F_0$ change of amplitude (DETQ red, vehicle black; mean ± SEM; $n = 4$ mice, 16, 12, 7 independent channels for NAc,

CPu and PFC respectively; ***$p = 4.6347 \times 10^{-06}$ for NAc, ***$p = 1.0976 \times 10^{-04}$ for CPu **$p = 1.0976 \times 10^{-04}$ for PFC, Wilcoxon Rank Sum test, Bonferroni corrected). **e** DETQ decay times (DETQ red, vehicle black; mean ± SEM; $n = 4$ mice, 16, 12, 7 independent channels for NAc, CPu and PFC respectively; ***$p = 1.4388 \times 10^{-05}$ for NAc, ***$p = 1.0976 \times 10^{-04}$ for CPu, *$p = 0.0260$ for PFC, Wilcoxon Rank Sum test, Bonferroni corrected). **f** $\Delta F/F_0$ signal decomposed with the Wavelet transform to multiple temporal scales (3.3 to 20 seconds) (DETQ red, vehicle black; mean ± SEM). Bar (top) shows $p < 0.05$ two-sided Wilcoxon signed-rank test for the before/after injection comparison of temporal scales. Source data are provided as a Source Data file.

even the more sensitive next-generation versions (e.g. GRAB_DA3)[55]. Despite heated debate in the field about the nature and functions of tonic versus phasic DA release and the receptor mechanisms that detect them[1,56–59], these two distinct modes of DA release seem to be differentially involved in regulating specific brain functions[59–61]. On the one hand, tonic DA is necessary for motor control[62], and is involved in certain forms of synaptic plasticity[63], learning[61,64] and responses to drugs of abuse[65]. On the other hand, the role of phasic DA is mostly involved in reward learning by signaling prediction errors[40,66–69] or retrospective causal learning[70].

We demonstrate with either chemogenetic (hM4Di) or optogenetic (eNpHR) inhibition of VTA DA neurons that the DETQ-induced step-like increase in dLight1.3b signal in the NAc can be substantially reduced but not completely eliminated with transient inhibition of DA neuron firing, raising the intriguing possibility that both DA neuron activity-dependent and -independent factors contribute to establishing the tonic DA levels detected with our approach in the NAc. In the future researchers could use the approach to mechanistically test the cellular and/or molecular factors involved in determining the amounts of extracellular tonic DA across brain areas, for example, by probing the role of spontaneous, action potential-independent DA release mechanisms[62,71–73], with high spatiotemporal precision. Additionally, using this approach future studies could establish the relative changes in both tonic and phasic DA release during different motivational, physiological or disease states, or upon pharmacological stimulation with psychoactive drugs, and correlate them with downstream molecular signals (for example by using cyclic-AMP[74] or kinase activity reporters[75,76]). Since no

previously reported genetically encoded sensor or approach could provide the opportunity to simultaneously visualize both aspects of DA release in vivo, the adoption of DETQ into established DA imaging routines will catalyze a more thorough understanding of the complex and multifaceted nature of DA biology.

The effects observed through our in vivo photometry recordings using DETQ in mice echo the mechanism of action of DETQ on DRD1 in humans. In fact, previous work demonstrated the pharmacological effects of DETQ on behavior in transgenic mice carrying a human DRD1 in the place of the endogenous mouse receptor[24]. Our strategy might enable more detailed preclinical investigations into the pharmacological mechanisms underlying the therapeutic value of this important drug class and lead to a better mechanistic understanding of their effects for future clinical applications.

Another benefit of this approach is its lower likelihood of buffering endogenous DA signaling, which is a potential issue inherent to the use of all GPCR-based sensors. This important consideration is particularly relevant in conditions requiring high-level chronic expression of sensors in vivo[8]. Due to prolonged DA-sensor dissociation times, this risk is markedly higher for high-affinity genetically-encoded DA sensors[5]. In this regard, our approach offers an attractive option to reduce the overall risks of ligand buffering while maintaining access to measuring low-level DA release, given that dLight1.3b is a sensor with relatively lower affinity (hence lower likelihood of buffering effects, especially after chronic expression) and can be boosted on-demand. Finally, given that several PAMs and NAMs exist for multiple GPCRs, future studies could investigate whether a similar strategy could be deployed for other GPCR-based sensors for which these

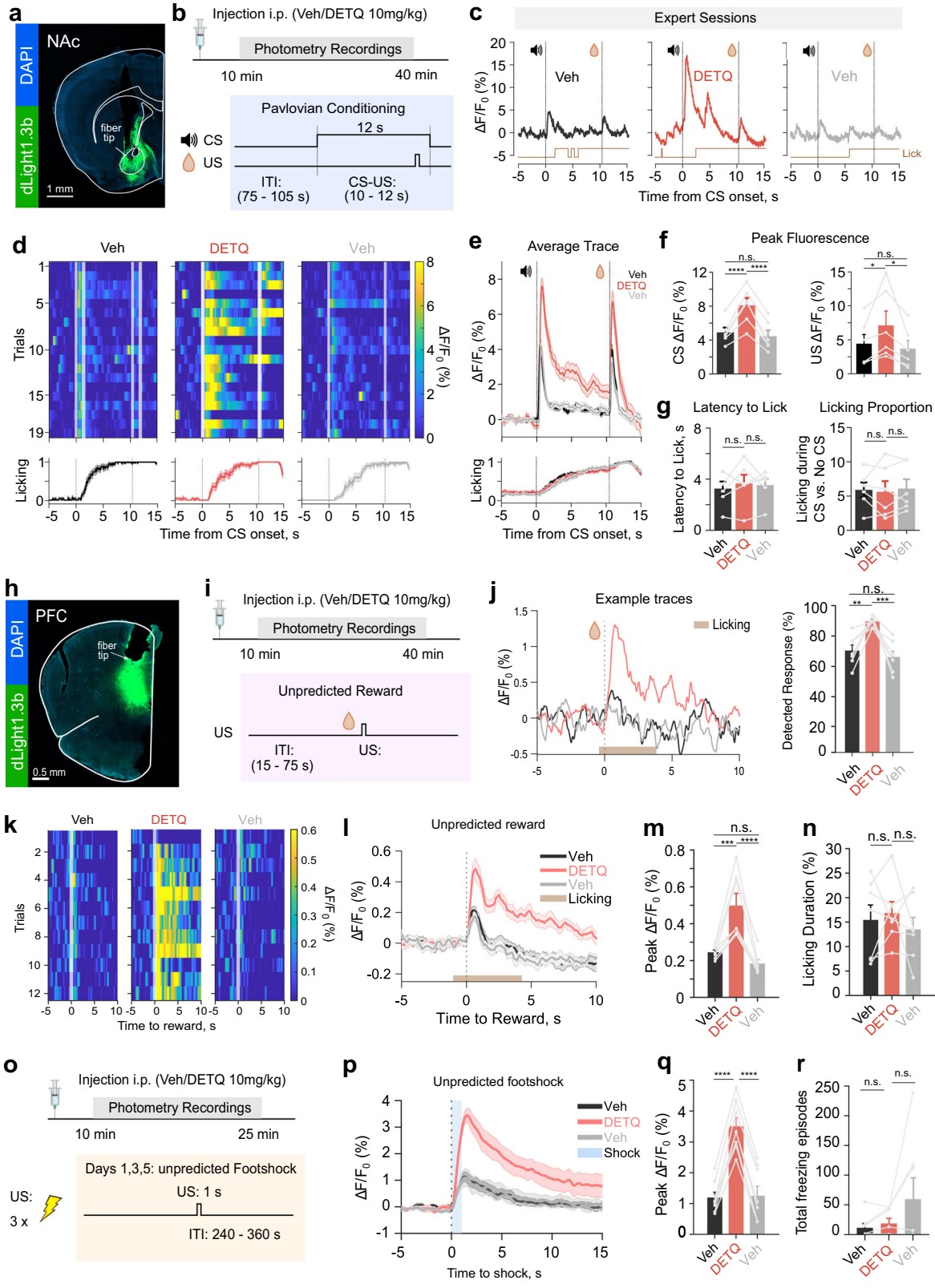

drugs exist[77], which could contribute to rapidly expanding the range of application of these tools via affinity tuning.

## Methods

### Ethical statement

All animal procedures were performed in accordance to the Animal Welfare Ordinance (TSchV 455.1) of the Swiss Federal Food Safety and Veterinary Office and were approved by the Zurich Cantonal Veterinary Office, the animal ethics committee of the Université de Montréal, or other relevant National and Institutional regulatory bodies.

### Molecular cloning

Sensor mutants were obtained by site-directed mutagenesis with custom-designed primers (Thermo Fisher) starting from a dLight1.1

**Fig. 8 | DETQ enhances photometry-based DA detection during naturalistic behaviors. a** NAc dLight recordings. **b** Experimental timeline and Pavlovian protocol. **c** Example dLight1.3b traces and licking responses (expert sessions) after Veh or DETQ. Dashed lines: 12-s tone or reward delivery. **d** Top, representative heatmap showing trial-to-trial variability of CS-evoked dLight1.3b response after Veh/DETQ. Bottom, average licking profile. **e** Average dLight1.3b response and average licking profile ($n = 6$ mice). **f** Quantification of peak CS- and US-evoked ΔF/F0 (CS: ****$p = 3.2 \times 10^{-5}$ for Veh day 1 vs. DETQ, ****$p = 1.0 \times 10^{-5}$, for DETQ vs Veh day 2, $p = 0.52$ for Veh day 1 vs. day 2; US: US: *$p = 0.044$ for Veh day 1 vs. DETQ, *$p = 0.012$, for DETQ vs Veh day 2, $p = 0.72$ for Veh day 1 vs. 2 ($n = 6$ mice). **g** Left, no difference in latency of anticipatory licking upon CS-detection between Veh/DETQ (Latency to lick: $p = 0.69$ for Veh day 1 vs. DETQ, $p = 0.96$ for DETQ vs. Veh day 2, $p = 0.83$ for Veh day1 vs. 2, $n = 6$ mice). Right, 6-times higher licking during CS vs. no CS; no difference between Veh/DETQ (Lick proportion: $p = 0.93$ for Veh day 1 vs. DETQ, $p > 0.83$ for DETQ vs Veh day 2, $p = 0.97$ for Veh day1 vs. 2 $n = 6$ mice). **h** PFC dLight recordings. **i** Experimental timeline and unpredicted reward protocol. **j**, Left: Example dLight1.3b traces and average licking responses from all trials (brown-shaded area). Right: response to reward detected in $70.62 \pm 3.93\%$ of single trials

after Veh day 1, $90.91 \pm 1.40\%$ for DETQ and $66.15 \pm 4.03\%$ for Veh day 2 (**$p = 4.2 \times 10^{-3}$ for Veh day 1 vs. DETQ, ***$p = 9.00 \times 10^{-4}$, for DETQ vs. Veh day 2, $p = 0.65$ for Veh day 1 vs. 2). **k** Representative heatmap showing trial-to-trial variability of US-evoked dLight1.3b response after Veh/DETQ. **l** Average dLight1.3b response and average licking profile ($n = 7$ mice). **m**, Quantification of peak Unpredicted Reward-evoked ΔF/F0 (***$p = 5.25 \times 10^{-4}$ for Veh day 1 vs. DETQ, ****$p = 7.4 \times 10^{-5}$, for DETQ vs. Veh day 2, $p = 0.43$ for Veh day 1 vs. 2) ($n = 7$ mice). **n**, No difference in lick duration associated with reward collection (top: $p = 0.79$ for Veh day 1 vs. DETQ, $p = 0.91$ for DETQ vs. Veh day 2, $p = 0.96$ for Veh day 1 vs. 2). **o**, Unpredicted shock protocol. **p** Average dLight1.3b response ($n = 8$ mice). Blue-shaded area: unpredicted shock. **q** Quantification of peak Unpredicted Shock-evoked ΔF/F0 (****$p = 3.5 \times 10^{-9}$ for Veh day 1 vs. DETQ, ****$p = 4.9 \times 10^{-9}$, for DETQ vs. Veh day 2, $p = 0.94$ for Veh day 1 vs. 2) ($n = 8$ mice). **r** No difference in total freezing episodes associated with footshock ($p = 0.99$ for Veh day 1 vs. DETQ, $p = 0.99$ for DETQ vs. Veh day 2, $n = 8$ mice). One-way ANOVA and Tukey or Sidak, or Friedman's test by Dunn's tests used. Data shown as mean/SEM. Graphics in **b, i, o** were created with BioRender.com. Source data is provided.

template (Addgene #111053). PCR reactions were performed using a Pfu-Ultra II Fusion High Fidelity DNA Polymerase (Agilent). For cloning dLight1.3b and AlloLite-ctr into viral vector plasmids, BamHI and HindIII restriction sites were added flanking the sensor coding sequences by PCR-amplification, followed by restriction cloning into pAAV-hSynapsin1-WPRE, obtained from the Viral Vector Facility of the University of Zurich. DNA sequences encoding the mouse DA receptors D1 (msDRD1), D2 (msDRD2) and D5 (msDRD5), the mouse alpha-1A adrenergic receptor (msADRA1a), the mouse beta-2 adrenergic receptor (msβ2AR) and the mouse 5-hydroxytryptamine receptor 6 (ms5HT6) were ordered as gene fragment (Twist Bioscience) or were previously reported[27], and were cloned into a mammalian expression vector downstream of a CMV-promoter (Addgene plasmid catalog no. 60360) by restriction enzyme cloning using NotI and HindIII sites. To generate C-terminal fusions of SmBiT onto the different receptors, the SmBiT sequence was PCR-amplified from pCMV-nLightG-SmBiT plasmid using custom-designed primers (ThermoFisher Scientific) and PfuUltra II Hotstart PCR Maser Mix (Agilent). The sequence was subsequently cloned in place by Gibson Assembly using the NEBuilder HiFi DNA Assembly Master Mix (New England Biolabs). All sequences were verified using Sanger sequencing (Microsynth).

### Cell culture, imaging, and quantification

HEK293T cells (ATCC #CRL-3216) were cultured in DMEM medium (Thermo Fisher) supplemented with 10% FBS (Thermo Fisher) and 100 μg/ml Penicillin-Streptomycin mix (Thermo Fisher) and incubated at 37 °C with 5% $CO_2$. Cells were transfected at 50–60% confluency in glass-bottomed dishes (either individually or in 24-well plates) using the Effectene transfection kit (Qiagen) according to manufacturer instructions, and imaged 24–48 h after transfection. The Flp-In™ T-REx™ 293 cells (Thermo Fisher) with a stably integrated dLight1.3b cassette[48] were cultured as follows: the cells were cultured using HEK293T cell medium with the addition of Blasticidin (15 μg/ml) and Hygromycin B (200 μg/ml). To induce the expression of dLight1.3b, doxycycline (500 ng/ml) was added to the culture medium for 24–48 hours. Before imaging, cells were rinsed with 1 mL of Hank's Balanced Salt Solution (HBSS, Life Technologies) supplemented with $Ca^{2+}$ (2 mM) and $Mg^{2+}$ (1 mM), and imaged at room temperature with a final HBSS volume of either 100 μL for each individual 1.5 cm glass bottom dish, or 500 μL for 24-well plates. Time-lapse imaging was performed on an inverted Zeiss LSM 800 confocal microscope using a 488 nm laser and a 40X oil-based objective. During imaging, ligands were added in bolus on the cells using a micropipette to reach the final specified concentrations of ligand over the cells. For quantification, except otherwise noted, regions of interest (ROI) were selected manually using the threshold function of Fiji to isolate the cell

membrane. Sensor response (ΔF/F) was calculated as the following: (F(t)-F_base)/F_base with F(t) being the ROI fluorescence value at each time point (t), and F_base being the mean fluorescence of the 10-time points immediately prior to ligand addition.

### Nanoluciferase complementation-based assay

To measure the effect of DETQ on mini-G protein recruitment by different GPCRs we used a Nanoluciferase (NanoBit)-based complementation assay[26,27]. For this HEK293T cells were seeded in six-well plates (Techno Plastic Products) at a density of 300,000 cells per well and co-transfected with plasmids encoding either hmDRD1, msDRD1, msDRD2, msDRD5, msADRA1a, msβ2AR or ms5HT6 carrying a C-terminally fused SmBiT, in combination with the corresponding mini-G protein probe[78] (mini-Gs: for DRD1, DRD5, β2AR, 5HT6R; mini-Gq: for ADRA1a; or mini-Gi: for DRD2), which were N-terminally fused to LgBiT, in a 1:1 DNA ratio using Effectene transfection reagent (Qiagen). 24 h after transfection cells were detached using Versene (ThermoFisher Scientific), washed, centrifuged (RT, 150 g, 3 min), and resuspended ($0.5 \times 10^6$ cells ml⁻¹) in Fluorobrite DMEM (ThermoFisher Scientific) complemented with 30 mM HEPES (ThermoFisher Scientific). First, 95 μl of the cell suspension were added into each well of a white flat-bottom 96-well OptiPlate (PerkinElmer), then 25 μl of a 20-fold dilution of Nano-Glo reagent in LCS buffer (Promega) and last either 5 ul of pure Fluorobrite-DMEM or 5 ul of DETQ diluted in Fluorobrite DMEM (final concentration 100 nM for DRD1; 1 μM for all other receptors). The plate was incubated at 37 °C for 30 min, before the luminescence signal was measured at 37 °C using a Tecan Spark plate reader before and after the manual addition of 25 μl of either ligand solution (final concentrations ranging from 1 nM to 100 μM), pure Fluorobrite-DMEM or 25 μl of a DETQ solution (1 μM final concentration).

### Primary DA neuron culture and imaging

Primary DA (DA) neurons were grown on a layer of cortical astrocytes. The astrocytes were first prepared from P0-P2 C57BL6 mice, grown in T75 flask and then seeded on collagen/poly-L-lysine coated 15 mm glass coverslips at a concentration of 100,000 cells/ml. The cells were allowed to grow until close to confluence before the seeding of neurons. Primary DArgic neurons were prepared and cultured from P0-P2 mouse pups using a previously described protocol[79]. Briefly, the ventral midbrain of P0-P2 pups was dissected from DAT^IRES cre x Ai9 mice in which DA neurons expressed the red fluorescent protein tdTomato. These were produced by crossing DAT^IRES cre mice (catalog #006660, Jackson Labs[31]) to Ai9/tdTomato mice (catalog #007905, Jackson Labs[32]). Both of these lines were on a C57BL/6 J background. Midbrain cells were seeded at a concentration of 240,000 cells/ml on top of the

astrocyte monolayer and allowed to grow for 10 days (10 DIV). Two days before imaging, expression of dLight1.3b was induced in stable HEK cells with doxycycline. One day prior to imaging, HEK cells were lifted using trypsin, and 150,000 HEK cells were seeded on top of 10 DIV-cultured neurons in doxycycline-supplemented neuronal medium. Imaging was performed on a confocal microscope (Nikon A1 HD25) using a 40X dry objective and a 488 nm laser to excite dLight1.3b and a 546 nm laser to activate tdTomato and identify DA neurons. On the day of imaging, physiological saline containing drugs at the specified concentration (AP-4, Sigma; SCH23390, Sigma; DETQ, Enamine) were applied on the culture using a computer-controlled perfusion system (Automate system). A field containing one or more DA neurons was selected based on the tdTomato signal. Images were acquired using the confocal system resonant scanner mode at an interval of 600 ms.

Event detection was performed using a thresholding algorithm adapted from[80]. In brief, the recorded field of view was divided into square ROIs of 55 um x 55 um size. Mean intensity across the entire duration of recording was obtained for each ROI and subsequently smoothed with a 6 seconds rolling window. To account for the effect of global fluorescent increase caused by large DA release events, an influence factor of 0.2 per frame was used to adjust the smoothed mean intensity value. A rolling standard deviation was calculated with the same window size. Events were defined as timepoints at which the fluorescence intensity reading exceeds the averaged intensity by 5 standard deviations.

### Virus production

The AAV vectors were produced by the Viral Vector Facility of the University and ETH Zürich (VVF). All other viruses were either obtained from the VVF repository or from the University of Pennsylvania Vector Core. The viral titer of the viruses used in this study were: AAV9.CAG.AlloLite-ctr, $6.0 \times 10^{12}$ GC/mL; AAV9.CAG.dLight1.3b, $6.6 \times 10^{12}$ GC/mL; AAV9.hSyn.dLight1.3b, $7.9 \times 10^{12}$ GC/mL; AAV5.hSyn.Flex.ChrimsonR-tdTomato, $5.2 \times 10^{12}$ GC/mL; AAV5-hEF1α-dlox-hChR2(H134R)-EYFP, $4.7 \times 10^{12}$ vg/mL.

### D1-PAM drug preparation

DETQ [2-(2,6-dichlorophenyl)-1-((1S,3R)-3-(hydroxymethyl)-5-(2-hydroxypropan-2-yl)-1-methyl-3,4-dihydroisoquinolin-2(1H)-yl)ethan-1-one] was either synthetized from Eli Lilly[25] (batch: BK7-E16933-083-3, Indianapolis, IN, USA) or purchased through Enamine (batch: EN300-657189, Riga, Latvia). Note that DETQ is available at multiple other commercial vendors (CAS No. 1638667-81-8). Both DETQ batches were equally validated in vitro in HEK293T cells. DETQ from Eli Lilly was utilized for in vitro experiments on HEK293T cells and for in vivo single-fiber photometry and optogenetic experiments. DETQ from Enamine was utilized for in vitro experiments with cultured DA neurons, as well as ex vivo two-photon imaging and in vivo multi-fiber photometry recordings. DETQ was dissolved in DMSO for in vitro and ex vivo experiments, then diluted to the appropriate concentration in HBSS (in vitro) or ACSF (ex vivo). For in vivo experiments, DETQ was dissolved via ultrasonication in the vehicle consisting of 20% 2-hydroxypropyl-betacyclodextrin (HPBCD; prepared in sterile milliQ water, Sigma: PHR1440). Solution for injection consisted of a fine suspension, prepared freshly daily, mixed prior to use and injected intraperitoneally (i.p.) using 26 G Sterican needles at a dose of 10 mg/kg (except for dose-response experiments).

### Animals

Mouse pups from DAT$^{IRES}$cre (B6.SJL-Slc6a3tm1.1(cre)Bkmn/J; Jackson Labs)[31] x Ai9 (B6;129S6-Gt(ROSA)26Sor tm14(CAG-tdTomato)Hze /J; Jackson Labs)[32] mice were used for preparing primary DA neuron cultures. Adult DAT$^{IRES}$cre (B6.SJL-Slc6a3tm1.1(cre)Bkmn/J; Jackson Labs)[31], DRD1:cre (B6;129-Tg(Drd1-cre)120Mxu/Mmjax;Jackson Labs)[81], or wild-type (>8 weeks old) C57/Bl6J mice of both sexes were used in this study. Mice were kept with *ad libitum* access to chow and water on

either normal or reversed 12-h/12-h light/dark cycle for two-photon or photometry/optogenetic experiments, respectively. Behavior experiments were performed during the dark phase.

### In vivo pharmacokinetic experiments

To measure free brain and plasma concentrations of DETQ in the mouse, male C57Bl6/N mice from Taconic Biosciences (Germantown, NY, USA) weighing 23–25 g (9–10 weeks old) were used for this study. The animals were acclimated for at least 72 hours after arrival from the breeder and housed five per cage in "shoe box" size cages with free access to food and water (lights on from 5 am to 5 pm) before use. The experiment was approved by the Institutional IACUC committee, and Eli Lilly is an AAALAC-certified facility. DETQ (Eli Lilly, see above) was formulated as a fine suspension in 20% HPBCD in purified water using ultrasonication. The drug was administered intraperitoneally at 10 mg/kg body weight (10 mL/kg. dose volume) with four animals per time point (0.25, 0.5, 0.75, 1, 1.5, and 2 hr). The animals were decapitated at the time points indicated above, and their frontal brains (mainly frontal cortex) were rapidly dissected and frozen. Blood was collected at decapitation, and plasma was separated by centrifugation. Both samples were then stored at −80 °C until further analysis. DETQ concentrations were measured in plasma and brain tissues by liquid chromatography with tandem mass spectrometry (LC/MS). The samples were analyzed using an AB Sciex API 4000 quadrupole mass spectrometer (Applied Biosystems, Foster City, CA) with a TurboIon-Spray interface. Brain samples were homogenized in a methanol-water solution (20:80) using a probe sonicator. Both brain and plasma samples underwent extraction procedures. Calibration curves were established by spiking blank plasma or brain samples with an internal standard. The system utilized Shimadzu LC-10AD pumps controlled by an SCL-10A controller (Kyoto, Japan). Sample injections were performed by a Z-215 liquid handler (Gilson, Middleton, WI). Chromatographic separation of all compounds was achieved using a Betasil javelin C18 HPLC column, 20 mm × 2.1 mm, 5 μm (Thermo Fisher Scientific, Waltham, MA). The unbound fraction in brain tissue samples was determined by equilibrium dialysis in mouse brain homogenate mixtures after incubation of the compound for 4.5 hours[82]. The unbound brain concentrations were calculated from the following equation[24]:

$$\text{Total brain concentration} * \text{Unbound fraction in brain} = \text{Unbound brain concentration}(Cu, brain)$$

### Animal surgeries and stereotaxic viral injections

For electrophysiological recordings in the dorsal striatum, DRD1:cre mice were injected unilaterally with AAV5-hSyn-dlox-ChrimsonR-tdTomato[33] at a 1:2 dilution into the DS (AP + 0.5, ML ± 1.6, DV −3.35, 300 nL volume). For two-photon imaging in acute brain slices, wild-type mice were unilaterally injected with AAV9-Syn-dLight1.3b (3 × $10^{12}$ GC/mL) into the striatum (300 nL) and medial prefrontal cortex (200 nL) using the following coordinates (relative to bregma, mm): striatum (AP + 1.2, ML + 1.7, DV −3.1); prefrontal cortex (AP + 2.4, ML + 0.5, DV −1.8). For labeling VTA projections to PFC, DAT$^{IRES}$cre mice were injected with a cre-dependent AAV expressing ChR2 (AAV5-hEF1α-dlox-hChR2(H134R)-EYFP in the VTA (AP −3.2, ML + 0.4, DV −4.2, 900 uL). For simultaneous optogenetic or chemogenetic stimulation of VTA terminals and photometry imaging, DAT$^{IRES}$cre mice were injected unilaterally with a cre-dependent AAV5-hSyn-flex-ChrimsonR-TdTomato, AAV5.hEF1a.Flex.eNpHR3.0.RFP or AAV5.hSyn.Flex.hM4D.mCherry into the medial VTA (AP −3.2, ML + 0.4, DV −4.2, 600 nL) and with AAV9-CAG-dLight1.3b, or AAV9-CAG-AlloLite-ctr in either the PFC (AP + 1.9, ML + 0.40, DV −2.4, 600 nL volume) or in the NAc core (AP + 1.3, ML + 1.35, DV −4.5, 600 nL volume), as specified in the figure legends. For photometry

recordings of behaviorally-evoked DA release in NAc or PFC, wild-type mice were injected unilaterally with undiluted AAV9-CAG-dLight1.3b, or AAV9-CAG-AlloLite-ctr in the PFC or in the NAc using the same coordinates as above. For measuring activity of DRD1 neurons, DRD1-cre mice were injected unilaterally with a cre-dependent AAV5-hSyn-Flex-jGCAMP8m into the NAc shell (AP + 1.5, ML 0.5, DV −4.4 or AP + 1.0, ML 0.6, DV −4.65, 230 nL). Low-autofluorescence optic fibers were either assembled with an optical fiber (0.48 NA, 400 μm core) (ThorLabs) secured into a ceramic ferrule (1.25 mm or 2.5 mm OD, ThorLabs) or were purchased from Doric Lenses and were implanted in the same surgery approximately 100–200 μm above the injected region. For multi-site photometry recordings, AAV9-CAG-dLight1.3b ($6.6 \times 10^{12}$ GC/mL) was injected via glass capillaries. Mice were anesthetized with 2% isoflurane (mixed in pure oxygen) and their body temperature maintained at 37 °C with a heating pad. To prevent inflammation and for analgesia, we applied Metacam (s.c., 0.1 μl/g bw). Small slit-like craniotomies were made to allow for virus injections and fiber-array implantations. Multi-fiber implantation was performed in the same surgical session. After the removal of connective tissue, the skull was dried and iBond (Kulzer, Total Etch) was applied to ensure adhesion of the skull to the connective dental cement. First, ~120 nl of virus-containing solution was pressure-injected into all areas of interest at a rate of about 120 nl/min. Subregions were targeted among 3 areas: PFC (3 injections bregma 2/2/2, lateral 0.25/0.25/0.6, depth 2.3/1.3/0.5; distances in mm); NAc (5 injections 1.4/1.2/1, 1/1.25/1.6, depth 3.6/3.6/4.5); CPu (4 injections 0.7/0.5/0, 1.7/2/2.2, depth (3-2.5)/(3-2.5)/3.5). In order to allow for local diffusion and to minimize potential reflux, the glass injection pipette was kept in place for 6 min after each injection. The 12-fiber-array was implanted with the help of a stereotaxic manipulator with fiber tips targeting the injection zones, 12-fiber-array was tilted at an angle of 45 degrees relative to the midline. The fiber array was oriented such that the most lateral fiber efficiently targeted CPu and the most medial fiber targeted PFC. Dental cement was applied (Tetric EvoFlow A1) on the skull around the implants, followed by UV light curing.

## Fast-scan cyclic voltammetry

Striatal DA release was measured using fast-scan cyclic voltammetry (FSCV) with a carbon fiber electrode (CFE; diameter 7 μm, length 50–100 μm) positioned approximately 100 μm below the surface of the slice. The CFE was scanned across a triangular waveform (−0.4 to +1.3 V, 400 V/s) at 10 Hz, and DA release was evoked by single-pulse electrical stimulation (30 μA, 0.5 ms) with an inter-stimulus interval of 180 s. DA release was evoked in the presence of DHBE (1 μM) to block nicotinic acetylcholine receptors. Peak background-subtracted DA oxidation current was measured at +0.6 V, and drugs were washed on after establishing a stable baseline.

## Electrophysiology in acute brain slices

For experiments involving SKF38393 application, to identify DRD1 receptor (DRD1)-positive neurons in the dorsal striatum, DRD1:cre mice injected with AAV5-hSyn-dlox-ChrimsonR-tdTomato were used. 3–4 weeks after viral injection, mice were anesthetized with 5% isoflurane and euthanized by decapitation. Coronal 280 μm slices containing DS were prepared using a vibrating blade microtome (HM 650 V, Thermo Fisher Scientific), in ice-cooled artificial cerebrospinal fluid (ACSF) containing (in mM): NaCl 120, KCl 2.5, MgCl₂ 1.0, CaCl₂ 2.5, Na₂HPO₄ 1.25, NaHCO₃ 26.0, glucose 14.6 and HEPES 5.0 (Osmolarity: 305–310 mOsm/kg), bubbled with 95% $O_2$ and 5% $CO_2$. Slices were kept at 34 °C for 10–20 minutes and then at room temperature until recordings. In the recording chamber, slices were superfused with ACSF at 30–32 °C. Patch pipettes (3–7 MΩ) were pulled from borosilicate glass pipettes (GC150T-10, Harvard Apparatus). The internal solution contained (in mM): potassium gluconate 130, MgCl₂ 4, MgATP

3.4, Na₃GTP 0.1, creatine phosphate 10, HEPES 5, and EGTA 1.1. The membrane potential was measured using EPC 10 USB Patch Clamp Amplifiers (HEKA). The access resistance was monitored by a hyperpolarizing step of −10 mV. The identification of DRD1-positive neurons was performed by the presence of red fluorescence protein and photo-evoked current or potential (@2 Hz, 10 ms duration, 2 pulses or 500 ms continuous pulse. light source: Colibri 7, Zeiss; 630 nm wavelength, 1 mW). To assess the excitability of neurons, positive currents were injected (0–500 pA) and the number of action potentials was counted at each current step. To measure input resistance, negative currents were injected (−200 to 0 pA, every 50 pA, 500 ms), as previously described[83]. The resting membrane potential was measured at 0 pA current input.

For experiments involving DA application, acute brain slices (240 μm) were prepared from 5-week-old, Drd1-tdTomato mice (Jackson Laboratories, #016204). DRD1-MSNs were identified by tdTomato fluorescence under 2 P microscopy (λ = 920 nm) and patched using an Axopatch 200B amplifier (Molecular Devices) with pipettes (3–5 MΩ) filled with (in mM): 135 K+ gluconate, 10 HEPES, 4 MgCl2, 1 mg/mL ATP, 0.1 mg/mL GTP and 1.5 mg/mL phosphocreatine (pH 7.3, 280 mOsm). Action potentials were evoked by current injection (50–500 pA, 500 ms) in current-clamped neurons. Neurons were recorded from in the presence of picrotoxin (100 μM), CGP55485 (0.3 μM), DNQX (10 μM), and MK801 (10 μM) to block GABAA, GABAB, AMPA and NMDA receptors, respectively. DA (10 μM) was applied to brain slices in the presence of cocaine (5 μM) to prevent reuptake.

## Two-photon imaging in acute brain slices

Coronal brain slices (240 μm) containing the striatum and PFC were prepared from AAV-injected mice after 3–4 weeks of viral expression. Briefly, mice were deeply anesthetized with isoflurane and transcardially perfused with ice-cold cutting solution containing (in mM): 75 NaCl, 50 sucrose, 6 MgCl₂, 2.5 KCl, 1.2 NaH₂PO₄, 0.1 CaCl₂, 25 NaHCO₃, and 2.5 D-glucose (bubbled with 5% $CO_2$ in $O_2$). The brain was quickly removed and sectioned using a vibrating blade microtome (VT1200S; Leica) before slices were transferred to an incubation chamber filled with artificial cerebrospinal fluid (ACSF) containing (in mM): 126 NaCl, 2.5 KCl, 1.2 MgCl₂, 1.2 NaH₂PO₄, 2.5 CaCl₂, 21.4 NaHCO₃, 11.1 D-glucose and 10 μM MK-801 (34 °C, bubbled with 5% $CO_2$ in $O_2$, pH 7.4). After 45 min equilibration, slices were individually transferred to a recording chamber for 2 P imaging. NMDA, AMPA, GABA-A, nicotinic, and muscarinic receptors were blocked using 10 μM MK801, 10 μM DNQX, 100 μM picrotoxin, 1 μM DHBE, and 0.3 μM scopolamine, respectively. Changes of dLight fluorescence were measured by 2 P laser scanning microscopy, using a custom-built BX51WI microscope (Olympus). Excitation (920 nm) was provided by a pulsed Ti:Sapphire laser (Chameleon Ultra I; Coherent) and scanned using a pair of X-Y galvanometer mirrors (6215; Cambridge Technology). Emission was collected by a water immersion objective (60x, N.A. 1.00; Olympus) and filtered (ET680sp and ET525/50m-2P; Chroma) before detection by GaAsP photomultiplier tube (PMT, H10770PA-40; Hamamatsu). The temporal kinetics of fluorescence changes were captured using fast acquisition speed (2 kHz), 'spot photometry' (Toronado; B.W. Strowbridge), by scanning the laser around a circular path (157 nm diameter) centered at the tip of the stimulating electrode or iontophoresis pipette which was visualized in the red channel (ET620/60 m; Chroma). The spatial dynamics of dLight fluorescence were captured by rasterized movie acquisition (30 × 40 μm, 3 Hz) and analyzed using ImageJ. Exogenous DA was applied by iontophoresis, using a thin-walled glass pipette filled with DA (1 M) and sulforhodamine 101 (300 μM), and ejected by short pulses of positive current ( + 20 to +200 nA, 5 ms) with a negative retention current to prevent DA leakage (−10 to −15 nA). Endogenous DA release was evoked by electrical stimulation using a double-tipped electrode pulled from septum theta glass capillaries (WPI), visualized with sulforhodamine 101 (10 μM). A single pulse (10 μA, 0.5 ms) was

used to evoke DA release in striatal slices, while a train of stimuli (5 × 20 μA at 25 Hz) was used in cortical slices. Changes of fluorescence are expressed as ΔF/F$_0$, wherein baseline fluorescence (F$_0$) was measured from the 300 ms before DA stimulation in spot photometry measurements, or defined as the average of 10 frames preceding stimulation in rasterized movies. Peak ΔF/F$_0$ was calculated by maximum peak detection from the time period following stimulation ( ~ 500 ms) or from the rasterized frame immediately following stimulation. Data were analyzed in AxoGraph (J. Clements) and ImageJ.

## In vivo single-fiber photometry and optogenetics or chemogenetics

Fiber photometry combined with optogenetics was conducted using a RZ10X LUX-I/O Processor with integrated LEDs (Tucker-Davis Technologies) combined with a filter/mirror minicube set (FMC6_IE(400-410)_E1(467-497)_F1(507-554)_E2(566-586)_F2(600-680)_S) (Doric Lenses). The 405 nm LED was passed through a 400–410 nm bandpass filter, the 465 nm LED through a 467–497 nm GFP excitation filter, then both were coupled to a dichroic mirror to split excitation and emission lights. Low-autofluorescence patch cords (400 μm/0.57 NA, Doric) were attached to the optic fibers (400 μm/0.66 NA, Doric) on the mouse's head to collect fluorescence emissions. Signals were filtered through 500–540 nm GFP emission filters coupled to a photodetector on the RZ10X. Signals were sinusoidally modulated, using the TDT Synapse software at 210 and 330 Hz (405 and 465 nm, respectively) via a lock-in amplification detector, then demodulated on-line. Data were sampled at 1017.3 Hz. 405 and 465 nm power at the patch cord were set to 30–50 μW. For optogenetic stimulation, a 635 nm LED was passed through the 580–680 nm minicube F2 port and reached the mouse through the same unique optic fiber; the LED was driven at 1 to 3 mW. To determine the 'temporal window for stable imaging' in the PFC using dLight1.3b recordings, DAT$^{IRES}$cre mice expressing ChrimsonR in VTA DA axons and dLight1.3b in the PFC were injected i.p. with vehicle at 0 min, followed by DETQ (10 mg/kg) 25 min later. After each injection, PFC DA axons were optically (opto) activated across 5 trials (2 s, 20 Hz, 20-ms pulses) for the vehicle (20 min) and 19 trials for DETQ (1h05 min) every 5 min. DA release was measured with fiber photometry. In another session, the same mice were injected with Veh, then DETQ (10 mg/kg) 20 min later, then cocaine (20 mg/kg) 25 min later, and the same optogenetic protocol was applied (2 s, 20 Hz, 20-ms pulses every 5 min). For optogenetic validation of AlloLite-ctr in the NAc, DAT$^{IRES}$cre mice expressing ChrimsonR in VTA DA axons were injected i.p. with vehicle (Day 1) or DETQ (10 mg/kg) (Day 8) at 0 min, followed by cocaine (20 mg/kg) 20 min later. After each injection, NAc DA axons were optically (opto) activated across 3 trials (2 s, 20 Hz, 20-ms pulses) every 5 min and DA released measured with fiber photometry. For optogenetic inhibition of DA transients, combined with dLight1.3b photometry in the NAc, DAT$^{IRES}$cre mice expressing eNpHR3.0 in VTA DA axons were injected i.p. with vehicle at 0 min and DETQ (10 mg/kg) 30 min later. After each injection, NAc DA axons were optically (opto) inhibited (10 s, continuous trials) every 3 min (8 trials/condition) and DA released measured with fiber photometry. For chemogenetic inhibition of DA transients, combined with dLight1.3b photometry in the NAc, DAT$^{IRES}$cre mice expressing hM4D in VTA DA axons were injected i.p. with vehicle at 0 min, then DETQ (10 mg/kg) or vehicle 20 min later, then with the hM4D ligand JHU37160 (J60[42]) (0.2 mg/kg 'low-dose' or 1 mg/kg 'high-dose') or vehicle 10 min later. DA release was measured with fiber photometry.

## Behavioral analysis and single-fiber photometry recordings in the open field

For combined locomotion/photometry experiments in the open field (OF) in tethered mice, a 50 × 50 cm arena was used. Fiber photometry was conducted using a Doric photometry rig combined with a filter/mirror minicube set (Doric Lenses), as above. Signals were sinusoidally

modulated, using the Doric Neuroscience Studio v.6.1.2.0 software at 208 and 572 Hz (405 and 465 nm, respectively). Data were sampled at 122 Hz. 405 and 465 nm power at the patch cord were set to 30–50 μW. Mice were habituated to the chamber for 3 consecutive days (30 minutes per day). On experimental days, locomotor activity was video recorded and synchronized with fiber photometry signal recording. The baseline DA signal was recorded for 20 minutes, then the mouse received i.p. injection of vehicle or DETQ (10 mg/kg) in interleaved sessions. Following injection, the DA signal was recorded for an additional 90 minutes. The OF chamber was cleaned with ethanol and water between mice. Recording sessions were repeated at 48 hours intervals. Mouse behavior as well as photometry signals were analyzed with a custom-written MATLAB script. Mouse locomotion was computed using the body center point of the animal. The center zone of the arena was defined automatically during behavior data analysis as the 25 × 25 cm square with the central point aligned to the center of the open field. The distance moved and time spent in the center zone were quantified in 10 minutes time bins and as the total during the 90 minutes after ip injections. To test the dose-dependent effects of DETQ on locomotion in untethered mice, mice were injected i.p. with vehicle, DETQ (3, 10 or 30 mg/kg) or cocaine (20 mg/kg) while their locomotor activity was recorded in an OF arena (60 x 60 cm) for 45 min. Videos were recorded with a Logitech C270 camera and analyzed with DeepLabCut (version 2.1.8.2)[84] on an existing open field network[85]. Representative traces were generated with the Anymaze (v7.2) software (Stoelting Co.).

## In vivo single-fiber photometry repeated across weeks

To determine inter- and within-mouse variability of dLight1.3b responses to DETQ (repeated weekly), DAT$^{IRES}$cre mice expressing dLight1.3b in the NAc core were injected i.p. with vehicle at 0 min and DETQ (10 mg/kg) 10–15 min later, then recording lasted 50 min. The experiment was repeated once per week across 3 weeks.

## In vivo single-fiber photometry with food restriction

To determine how metabolic state modulates tonic DA levels, DAT$^{IRES}$cre mice expressing dLight1.3b in the NAc core were tested in either the fed state (ad libitum), the food-restricted state (mice food restricted to 90% of baseline body weight for 5–7 days), or after 4 weeks recovery from food-restriction (ad libitum). Daily food intake was measured per cage (2–3 mice per cage). Photometry data from 'ad lib fed week 1' is taken from the last week of the "photometry repeated across weeks" as the experimental design was the same and results showed highly stable responses within individual mice, making it a perfect baseline. Mice were injected i.p. with vehicle at 0 min and DETQ (10 mg/kg) 10–15 min later, then recording lasted 50 min.

## Pavlovian conditioning and unpredicted reward or shock

Cue-reward paradigm (NAc recordings) was carried out in an operant chamber within a sound-attenuating box with low red light illumination (30 Lux). Chamber functions were controlled by custom-written Matlab programs via a National Instrument board (NI USB 6001). After habituating mice to the experimental chamber (4 consecutive days for 5–10 min) mice were food restricted to 90% ad libitum body weight. During the habituation phase, in order to learn the position of the lick port mice received a milk reward (5 μL of 30 % condensed milk solution) at random time intervals (total 10 times, ITI = 10 - 25 s). During the next session mice were trained to initiate reward delivery via licking. Next, in cue-reward learning sessions, a CS (80 dB 5 kHz tone) was turned on for 12 seconds and was followed by US delivery (5 μL of 30 % condensed milk solution) with variable inter-trial intervals (ITI, randomly drawn from a uniform distribution between 75 and 105 seconds). During learning sessions, the delay between CS and US was increased every training day from 2 to 10 seconds until mice became expert. During expert sessions CS-US pairings were repeated

20 times per session (~30 min for each session). Mice underwent a total of 12 sessions for cue-reward learning. 10 minutes before each session drugs were injected i.p. (Veh on Days 1 and 5, DETQ 10 mg/kg on day 3 of Expert sessions). The rewards were delivered from 10 to 40 min post vehicle or DETQ injection. Chambers and lick ports were sanitized with ethanol between mice. For all sessions, lick data and photometry recordings were simultaneously obtained; licking behaviour was detected by a capacity-based sensor. Latency to reward-anticipatory licking was calculated as the time between the CS and lick onset. The lick proportion was calculated as the percent time mice spent licking during CS presentation divided by the percent time mice spent licking during the rest of the session (without CS). Similarly, for the unpredicted reward test (PFC recordings), mice were habituated to the chamber and trained to initiate US (5 µL of 30 % condensed milk solution) delivery via licking. To prevent mice from habituating to the reward, a delay of 48 h was set between recording sessions and a maximum of 12 trials per session were given (in random intervals from 15 to 75 s between reward availability). For unpredicted footshock drugs were injected i.p. (DAT$^{IRES}$cre mice: Veh on Days 1 and 5, DETQ 10 mg/kg on Day 3; DRD1-cre mice: Veh on Day 1, Veh or DETQ 10 mg/kg on Day 7, SCH 23390 0.1 mg/kg on Day 14). After i.p. injection, mice were placed into an operant chamber. 3–4 electric footshocks (0.4 mA for 1 second) were delivered 10 to 25 min post vehicle or DETQ with variable intervals (randomly chosen from a uniform distribution between 240 and 360 seconds or between 120 and 240 seconds) without predictive cues. Mouse freezing behavior was monitored with a video camera (C310 Logitech,10 frames per second) positioned above the recording chamber and controlled by the Synapse software. Freezing episodes were quantified using Anymaze (v7.2) software (Stoelting Co.). Freezing was defined as the complete cessation of all movement for at least 1 s during a 25-min testing duration. Total freezing episodes were calculated as the number of times the animal froze during the entire test session.

### Data analysis for in vivo single-fiber photometry

Fiber photometry recordings were analyzed using custom-written Matlab scripts. Data was 10x downsampled and low-pass filtered (1–3 Hz). Change in fluorescence, $\Delta F/F_0$ (%), was defined as $(F-F_0)/F_0 \times 100$, where F represents the fluorescent signal (465-nm) at each time point. $F_0$ was calculated by applying a least-squares linear fit (polyfit) to the 405-nm signal to align with the 465-nm signal. For long-term recordings during optogenetic stimulation or chemogenetic inhibition linear fit was calculated across the entire recording or across the vehicle epoch; for event-related experiments fit was performed on fluorescent data obtained in the −60 to +60 s around opto events or −5 to −1 s around behavioral events. Event-related $\Delta F/F_0$ was baseline corrected against the $\Delta F/F0$ average in the −10 to −1 s timewindow before event onset. DA event-related responses were quantified by computing the value at the transient's peak value and the AUC in the 0 to 10 s or 0 to 60 s period post-event onset. For analysis of optogenetic stimulation, opto-evoked responses obtained with DETQ were normalized against the average response's peak value from opto-evoked trials obtained with vehicle. The maximal peak value of Veh-normalized $\Delta F/F_0$ after opto-onset was plotted over time and the plateau of maximal response was determined by averaging peak opto responses 5, 10, 15 and 20 min post-DETQ. We then looked at trials between 5 and 65 min post-DETQ to determine which trials belonged to a 'temporal window for stable imaging'. Trials were included if their peak responses were within 15% of the plateau response. We also calculated the 'temporal window of superior sensitivity', defined as the temporal window during which the opto-induced responses were significantly superior to the baseline state without DETQ (i.e. at 0 min post DETQ injection), by comparing the opto peak response at each trial to the 0 min trial. Opto-related transient off-decay time was computed using a double exponential fit applied to the average

opto-evoked responses starting from the timepoint at the response's peak. In the fitted curve, the time point where $\Delta F/F_0$ passed under 36.8% (i.e. 1/e) of the maximum intensity was selected as the transient off-decay time (i.e. mean lifetime) as in[86]. For event-related datasets, we also calculated the overall baseline $\Delta F/F_0$. Baseline $\Delta F/F_0$ was computed by first calculating $\Delta F/F_0$ on the entire dataset using the same polyfit calculation as above; then by averaging $\Delta F/F_0$ values in the −60 to −1 s before optogenetic trials. For chemogenetic experiments, weekly DETQ injection and food-restriction experiments, the baseline was set to zero by subtracting the average vehicle control fluorescence; and data smoothed across 10 sec chunks. Average $\Delta F/F_0$ in the 10–20 min post vehicle injection, 0–10 min post DETQ/Veh and 2–12 min post J60/Veh were computed. For weekly DETQ injection and food-restriction experiments, data was low-pass filtered (smooth) in 1 sec epochs. For weekly DETQ injection, AUC was calculated in the relevant drug epoch (Veh, DETQ) and normalized against the duration of that epoch (Veh: 10 min, DETQ: 50 min). For food restriction, AUC was calculated in the first 15 min after DETQ injection. For long-duration open field tests, in order to capture the bleaching of each trace over the long-term fiber photometry recording, the 405 and 465 nm excited signals were fitted with a polynomial function of 3$^{rd}$ degree using MATLABs polyfit function (Supplementary Fig. 10a). Each signal was then divided by its fit, in order to correct for bleaching and to normalize. Finally, $\Delta F/F_0$ was calculated as the difference between bleaching-corrected signals excited at 405 and 465 nm. The baseline fluorescence level (thick green line in Supplementary Fig. 10b) was estimated as the moving median filter over a 10,000 datapoints (82 seconds) window size. AUC of the baseline was calculated within the +5 to +40 minutes time window post ip injection. For detection of spontaneous DA peaks the calculated $\Delta F/F_0$ was smoothed with a 50 datapoints (410 milliseconds) moving window, and the built-in Matlab function *findpeaks* was applied, with the threshold for minimal peak prominence set to 0.5% $\Delta F/F_0$.

### Multi-fiber photometry recordings

We used a light emitting diode (LED, M470L5 Thorlabs) at the 470-nm central wavelength for excitation of dLight1.3b. To achieve stable continuous operation, LEDs were run at 80% of maximal output power corresponding to an excitation power of ~1 mW/mm$^2$ per fiber channel. LED light was collimated with the 16 mm condenser lens (ACL25416U-A, Thorlabs). A microscope objective (TL4x-SAP; Thorlabs) coupled excitation light into the respective array of 12-fibers (Sylex). A dichroic beamsplitter (F58-486 dual line, AHF) coupled the excitation light into the objective (TL4x-SAP, Thorlabs) and transmitted the fluorescence in the dLight1.3b emission spectral window. To separate fluorescence signals from residuals of the excitation light and to minimize auto-fluorescence generated in a broader spectral range we used an emission filter (525/50 nm, F37-516, AHF, for dLight1.3b). To image the end face of the fiber array onto the camera we used a 100-mm focal tube lens (TTL100-A, Thorlabs). The image was created at the back focal plane of the tube lens on the sCMOS sensor (Dhyana 400BSI V2, Tucsen Photonics). We used components of MT fiber connector technology: ferrules that can accommodate 12 fibers (part number 17185; US Conec, Hickory, NC) and alignment pins (part numbers 16741, 16742; US Conec). Optical fibers with polyimide coating (100-µm core diameter, 124-µm outer diameter, NA 0.22; UM 22-100, Thorlabs) were fitted into the ferrule. For targeting specific subsets of brain regions (medial to lateral PFC, NAc, CPu) fibers were aligned to the template defining the required implantation depths.

### Analysis of multi-fiber photometry data

Fluorescence signals were expressed as percentage $\Delta F/F_0$ relative to the fluorescence baseline. The fluorescence baseline was defined as the minimal level of the signal after subtracting the auto-fluorescence baseline, that is the level of fluorescence estimated prior to the mouse

connection. Because in our experiment we wanted to estimate the slow DETQ decay times, we fitted the second-order polynomial trend to the recorded fluorescence signals. Prior to fitting the recording was split to the -1 h trace before injection and -2 h trace after the injection. Each trace was independently fitted with the polynomial (using detrend function, MATLAB). While polynomial fitted to the trace before injection reflected photobleaching dynamics, the polynomial fitted after the injection reflected DETQ pharmacokinetic properties. In all measured traces we visually verified the quality of fit. We estimated the maximum of the trend line and compared it to the DETQ and vehicle injection. Similarly, we estimated the reduction of the signal to 1/e level relative to the maximum, reflecting the DETQ pharmacokinetic properties. To analyse the $\Delta F/F_0$ temporal dynamics at multiple temporal scales separately, we decomposed the fluorescence traces with the wavelet transform (cwt function, MATLAB). We separately analysed measurement traces (30 min before and 30 min after the injection). Importantly signal from the injection itself did not contribute to the analysis of temporal dynamics, because we removed 5 min during/post DETQ injection for the baseline fluorescence to reach the maximum. We also used polynomial fit (detrend) on both traces to remove the DETQ decay dynamics (down slope and contribution from photobleaching). After we compared across independent channels and mice contribution of various temporal scales (in the range of 1–30 seconds) to the recorded $\Delta F/F_0$. To compare the peak $\Delta F/F_0$ and DETQ decay times across different sessions (on the DETQ and vehicle days) we used non-parametric Wilcoxon Rank Sum test. To compare the temporal scales recorded before and after the DETQ injection, during the same session, we used a non-parametric Wilcoxon signed-rank test.

### Immunohistochemistry
Mice were transcardially perfused with ice-cold 4% paraformaldehyde (Sigma) in PBS under deep pentobarbital anesthesia (200 mg/kg, i.p., total volume 200 uL, diluted in NaCl). To visualize implant location, skulls were soaked into PFA for 2 days. Brains were harvested, post-fixed in PFA overnight and washed in PBS. Free-floating 50–60 μm coronal sections were cut using a freezing or non-freezing microtome (Leica). Co-localization analysis and confirmation of viral expression/implants: sections were washed, and incubated overnight at 4 °C with primary antibodies against GFP (1:1000, #A6455 Thermofisher), DsRed (1:500, #632392 Clontech), mCherry (1:500, #M11217, Molecular Probes or 1:1000, ab167453, abcam) and tyrosine hydroxylase (TH) (1:500, #22941 Immunostar) in 5% serum, 0.1–0.5% Triton X-100 PBS. Sections were then washed and incubated with corresponding fluorescent secondary antibodies (1:1000, Thermofisher or Jackson Immunoresearch) and DAPI (1:2000, 62248, Thermofisher) for 1 h at RT. Washed sections were then mounted on slides and coverslipped with Vectashield with/without DAPI (VectorLabs). Digital images were acquired using a Zeiss LSM 800 confocal microscope and processed with Fiji Image J or Zeiss Zen Lite software.

### Statistics and Reproducibility
For in vitro luminescence assays, ligand traces were divided by buffer traces, the area under the curve for each trace was calculated using GraphPad Prism, values were normalized and fitted as dose response curve. The change in luminescence ($\Delta L/L_0$) upon addition of different ligands was calculated as follows: $(L_t − L_0)/L_0$, with $L_t$ being the Luminescence at ten timepoints t immediately after ligand application, and $L_0$ being the luminescence of ten timepoints before ligand application. The statistical significance was determined using Welch's ANOVA followed by Dunnett's multiple comparison test (comparison of ligand or vehicle to ligand). For ex vivo imaging experiments, statistical analyses were performed in Prism (GraphPad) using independent samples t-test or nonparametric Mann-Whitney U-test when assumption of normality was violated (assessed using Schapiro-Wilk's test). Curve fitting

parameters were compared by extra sum-of-squares F test. For in vivo imaging experiments, statistical tests were paired and unpaired t-tests (two-tailed) and repeated-measures ANOVA for parametric data, while they were Wilcoxon rank sum tests or Mann-Whitney for non-parametric data. Sidak or Tukey post-hoc tests correcting for multiple comparisons were used in the case of significant interactions in ANOVA. All p-values are indicated in the Results section. Data are displayed as mean with standard error of the mean (SEM). No statistical methods were used to predetermine sample size. No data were excluded from the analyses. Group allocations used in this study were randomly assigned to animals and/or cultured cells. The Investigators were not blinded to allocation during experiments and outcome assessment.

### Reporting summary
Further information on research design is available in the Nature Portfolio Reporting Summary linked to this article.

### Data availability
DNA and protein sequences for the sensors developed in this study are available in Supplementary Note 1. DNA plasmids have been deposited at the UZH Viral Vector Facility (https://vvf.ethz.ch/). Viral vectors can be obtained either from the Patriarchi laboratory, or from the UZH Viral Vector Facility. The protein structure for dopamine-bound dopamine receptor 1 used in this work can be accessed via PDB ID 7LJD. Raw data is available at https://zenodo.org/doi/10.5281/zenodo.10932251 or by emailing the corresponding author. Source data are provided with this paper.

### Code availability
Custom MATLAB code used in this work is available on Github at the following link: https://github.com/PatriarchiLab/InVivoPhotometry[87].

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

## Acknowledgements

The results are part of a project that has received funding from the European Research Council (ERC) under the European Union's Horizon 2020 research and innovation program (grant agreement no. 891959 to TP). We also acknowledge funding from H2020-ICT (grant agreement: 101016787) (TF and TP), and the Swiss National Science Foundation (project grant no. 310030_196455 to TP; 310030L_212508 to TP and BW; postdoc return grant no. P4P4PB_191069 to MAL and ambizione grant no. PZ00P3_193430 to MAL), the National Institutes of Health (R01-DA35821 to CPF and F32-DA51135 to AGY), and the Canadian Institutes of Health Research (CIHR) (LET). This work was also supported by CNRS (contract UPR3212), the FRM ("Amorçage de Jeunes Equipes" AJE202110014579 to YS) and by ITI NeuroStra (YS) from the University of Strasbourg. We would like to thank Ulrik Gether (University of Copenhagen) for kindly providing us the Flp-In™ T-REx™ inducible cell complete system, as well as J-C Paterna and the Viral Vector Facility of the Neuroscience Center Zurich (ZNZ) for help with virus production. We would like to thank Prof. Hanns Uli Zeilhofer, Prof. Helmchen, Martin Wieckhorst, Harald Osswald and Alex Kramer for advice, and help with behavior setups.

## Author contributions

T.P. conceived and led the study, designed the experiments, analyzed data, supervised the work. M.A.L. helped with experimental design, performed surgeries and experiments and analyzed data for optogenetics+photometry, chemogenetics+photometry, weekly DETQ injections, food restriction and unpredicted shock, did open field experiments, and helped with 2-photon pilots and in vitro data analysis. M.W. designed, automated, performed and analyzed in vivo photometry experiments during Pavlovian conditioning, unpredicted reward and

open field, and performed related surgeries. Z.K. performed sensor screening and characterization in HEK293T cells, analyzed data from in vitro experiments, and helped with the analysis of mouse behavioral data. A.G. performed and analyzed in vitro luminescence experiments. R.B.C. provided support with in vitro experiments. M.H. performed and analyzed electrophysiology experiments. M.A.L., L.S.C., M.W., A.M.M. and K.O. performed I.H.C. for in vivo photometry experiments, under the supervision of T.P. K.O. helped with computational analysis of in vivo data. A.M.M. helped with surgeries and photometry for unpredicted shock experiments. R.D. performed and Z.K. and Xuehan Zhou analyzed in vitro D.A. imaging in DArgic neuronal cultures under the supervision of LET and TP. L.R., C.G. and B.W. provided critical input and infrastructural support. S.C. and T.F. provided critical input and reagents. A.G.Y. performed and analyzed ex vivo two-photon imaging, fast-scan voltammetry and electrophysiology experiments, under the supervision of C.P.F. Xin Zhou, J.K. and K.A.S. performed and analyzed in vivo P.K. data of DETQ. SCh and Y.S. performed multi-fiber photometry experiments and analysed respective in-vivo data. TP wrote the manuscript with contributions from all authors.

## Competing interests

T.P. is a co-inventor on a patent application (PCT/US17/62993) related to the genetically encoded sensor technology described in this article. Xin Zhou, J.K. and K.A.S. are employees of Eli Lilly and company. All other authors have no competing interests.
