## [Peer Review File · Nature Communications]

REVIEWER COMMENTS

Reviewer #1 (Remarks to the Author):

From a chemical biology perspective I had no concerns--I reviewed only the in vitro pharmacology which appears to have been done and analyzed correctly.

Reviewer #2 (Remarks to the Author):

This is a nice methods paper that described a clever use of a positive allosteric modulator (DETQ) that shows species selectivity for human D1 receptors, together with an existing genetically encoded sensor for dopamine (DA), dLight1.3b, that is based on the human D1 receptor. The studies reported show increased sensitivity for DA detection using a range of preparation from cultured DA neurons to in vivo imaging of DA release during naturalistic behaviors. The authors propose that the use of DETQ is somewhat analogous to the introduction of DREADDs, in that this drug can be used without interfering with animal behavior. There are a few limitations of the approach, and some concerns, however. First, the in vitro experiments suggest an 8-fold increase in sensitivity for DA, but the conclusion after the behavior studies is that this is a 2-fold increase, which is helpful, but not ground-breaking. The second limitation is that the amplification shifts continuously after DETQ injection. Inter-animal experiments are challenging with dLight because of variable injection conditions/expression between animals, and this would add an additional requirement of precise timing to make sure a larger/smaller response was not due to recording at a slightly different time point. Although the short duration is noted as a caveat, the implications of this are not. There were also no experiments conducted to confirm that the monitored, amplified signal is DA. dLight has been characterized, but this needs to be done for the response with DETQ. Other concerns are outlined below.

1) The rationale for increasing DA sensitivity needs to be strengthened, given previous evidence that stimulated increases in [DA] are 10-30 μM (Patriarchi et al 2018).

2) In this light, the stimulated increases in [DA] reported here (Fig. 3), should already be saturating given the concentration-response curves in Fig. 1b, but yet an increase in dF/F is seen with 200 nM DETQ. These results need to be explained, and the present and previous findings reconciled in the text.

3) 185-187, 218. Decay time has many contributing factors. Most commonly, a change in decay time is used to suggest a change in the activity of the DA transporter (DAT) or the response time of a probe. As written, the implication is that the longer decay time here reflects the probe's ability to detect falling levels of DA more sensitively, so that DA can be detected for a longer period. If this is the hypothesis, this should be stated explicitly, and evidence presented to show that DETQ does not alter DAT activity or dLight1.3b response time. Indeed, the result shown in Fig. 3a using iontophoresis is very like what one sees when uptake is inhibited (active uptake often prevents detection of exogenously applied DA in the striatum).

4) 226-228. The species selectivity of DETQ for human vs. mouse/rat D1 receptors needs to be clearly explained at the beginning. Without that, the lack of effect on behavior is surprising, as it implies a lack of role for DA in the behaviors examined, or at least the lack of a modulable role. Additionally, the authors do not demonstrate this selectivity for need to demonstrate that this is the case in their hands. They report that DETQ alone does not increase D1-MSN excitability in mouse striatal slices, which provided a critical control. However, more convincing would have been to assess whether DETQ altered the effect of DA on these cells. This should be tested.

5) 259-261. Using a control (AlloLite-ctr) that is relatively insensitive to DA (Fig. 1D) and insensitive to the amplification seen with DETQ to conclude that DETQ does not promote DA release seems circular in that AlloLite-ctr could not detect enhanced baseline release if it happened.

6) Confusingly, AlloLite is used to refer to dLight1.3b + DETQ in some places, whereas in other places dLight1.3b alone is used as the DETQ-sensitive form. dLight1.3b + DETQ is actually more accurate than the new term, which actually requires two components (and is technically the same as the original sensor). The idea is good in principle, but not in practice if the authors are not consistent; for example, in li. 260, they refer to AlloLite, but mean AlloLite-ctr, which makes a big difference.

7) Given the results in Figs. 3a,d (showing a many-fold difference for the lowest injection currents), how the % increases in baseline were calculated after DETQ administration needs to be described in the main text, as the starting baseline also appears to be set to zero in Fig. 6c,d.

8) The results in the initial sections of the report describing in vitro studies indicate an 8-fold increase in sensitivity to DA; however, the conclusion after the in vivo studies is that there is a 2-fold difference in signal. The reasons for this may come back to a concern raised earlier about the actual levels of DA detected under different conditions. This should be discussed.

9) A general practice for characterization of any sensor is to conduct experiments to test whether it detects what it is supposed to detect. Although dLight has been well-characterized, the enhanced signals with DETQ have not. This is a concern, particularly for in vivo use. Basic studies to test whether signals detected after DETQ is DA might include local infusion of TTX, or TTX in VTA/SNc to show activity dependence and DA neuron dependence. Such tests often also include pharmacological manipulations that alter DA release in a predictable way (e.g., after DA depletion with reserpine or amplification by cocaine). Here, cocaine was used to show that AlloLite-ctr could detect an amplified DA signal, but this same experiment was not done with AlloLite to show a comparable % increase and change in decay time between AlloLite and dLight1.3b.

Minor

45-47. Voltammetric methods have reasonable spatial resolution – but limited to one site at a time. This should be reworded.

56-57. Cholinergic regulation of axonal DA release has been recognized for at least 20 years, so is not a recent finding.

124. Should be DA/NE here – implies greater selectivity for NE when written NE/DA.

167-169. Please specify firing rate “during current injection”, given the well-known lack of firing of MSNs in slices.

167. Superfusion, not perfusion.

Generally not necessary to capitalize words for abbreviations, like dorsal striatum (DS) or ventral tegmental area (VTA).

Terminology should be consistent throughout, for example Allolite and AlloLite are both used, and D1-PAM interleaved with DETQ (if used for variety, use DETQ first in each data section).

Reviewer #3 (Remarks to the Author):

This manuscript describes the interaction between a D1 PAM and dLight1.3b. The application of DETQ a D1-receptor dependent positive allosteric modulator results in a shift in the concentration required to activate the fluorescence of the dopamine sensor dLight1.3b. The maximal increase in fluorescence is not affected by the PAM. Rather, there is an increase in the affinity of agonists for the sensor. The increased affinity is also responsible for a slowing of the decay following removal of the agonist. The manuscript describes a comprehensive series of experiments that demonstrate the potential utility of using this PAM.

Comments

1. The use of a PAM to modulate the fluorescence increase induced by the activation of the dopamine sensor is an interesting idea that could be useful for experiments aimed at discovering the sites of dopamine release. Beyond that it seems that the use of this method for behavioral experiments will be limited by the action of the PAM on endogenous receptors. As described in the literature, this compound has multiple actions. It is hard to imagine how this approach will be more advantageous than development of a sensor that has higher affinity and/or a larger dynamic range than dLight1.3b.

2. The experiments that were used to suggest that DETQ had no direct action on the D1-receptor or dLight1.3b in vitro are convincing. Experiments aimed at the determination of DETQ on endogenous receptors are not convincing. Experiments with the use of a single high concentration of the D1 agonist, SKF38393, on the firing of medium spiny neurons were flawed. Knowing that DETQ did not increase the maximum change in fluorescence it makes one wonder if it would have a similar action on endogenous receptors. Does it change the concentration response to the D1 agonist? Without concentration response curves to dopamine and/or SKF38393 in the absence and presence of DETQ the interpretation of the results is limited. The same could be said for the experiments with the use of cocaine. Does the sensitivity (dose response) to cocaine change in the presence of DETQ?

3. That GPCRs are complicated molecules even after the manipulations that result in the construction of a transmitter sensor is nicely demonstrated by this work. This approach could be valuable for studies aimed at the detection of sites of dopamine release in experiments with brain slices and potentially in some experiments in vivo, however the interpretation of behavioral experiments will be very limited without extensive controls because of the interaction of the PAM with endogenous receptors.

Reviewer #4 (Remarks to the Author):

Marie and his/her colleagues have developed AlloLite, a sensitive system for detecting dopamine levels that can dynamically respond to low concentrations of DA. As described in their manuscript, AlloLite demonstrates significant benefits, including a broad range of sensitivity to dopamine levels and the ability to distinguish between tonic and phasic dopamine release signals. However, there are still specific questions regarding its practical use and potential applications that I would like the authors to address.

1. Previous research has indicated that DETQ acts as a selective positive allosteric modulator (PAM) for the human D1 dopamine receptor (DRD1), exhibiting limited potency in rodents (see reference 17, for instance). The authors have extensively shown that DETQ does not modify mice's behavior in their experiments. However, these findings do not conclusively prove that DETQ's action is restricted to the D1 receptor. There remains the possibility that DETQ might also influence non-dopamine receptors, such as dopamine transporters or other proteins, potentially affecting synaptic dopamine levels. This, in turn, could interfere with the fluorescence signals detected by the AlloLite system. The manuscript would benefit from addressing how the study controls for or mitigates these potential sources of error.

2. Is there a risk of fluorescence quenching in the AlloLite sensor due to prolonged laser exposure? It would be beneficial for the manuscript to discuss how the decrease in fluorescence intensity, which may occur during the detection process, is accounted for and evaluated in terms of its impact on the experimental outcomes.

We would like to thank all Reviewers for their positive and constructive feedback on our manuscript. We performed additional experiments and provide detailed clarifications to address their concerns as detailed in the point-by-point responses below. The original Reviewer's comments are shown in *blue italics*. Cited text from the revised manuscript is shown here in *black italics*. We also marked in blue any major text changes in the revised version of our manuscript to enable easy identification.

Reviewer #1 (Remarks to the Author):

From a chemical biology perspective I had no concerns--I reviewed only the in vitro pharmacology which appears to have been done and analyzed correctly.

We thank the Reviewer for the kind comments on our manuscript.

Reviewer #2 (Remarks to the Author):

This is a nice methods paper that described a clever use of a positive allosteric modulator (DETQ) that shows species selectivity for human D1 receptors, together with an existing genetically encoded sensor for dopamine (DA), dLight1.3b, that is based on the human D1 receptor. The studies reported show increased sensitivity for DA detection using a range of preparation from cultured DA neurons to in vivo imaging of DA release during naturalistic behaviors. The authors propose that the use of DETQ is somewhat analogous to the introduction of DREADDs, in that this drug can be used without interfering with animal behavior. There are a few limitations of the approach, and some concerns, however. First, the in vitro experiments suggest an 8-fold increase in sensitivity for DA, but the conclusion after the behavior studies is that this is a 2-fold increase, which is helpful, but not ground-breaking.

Please see our response to point 8 below.

The second limitation is that the amplification shifts continuously after DETQ injection. Inter-animal experiments are challenging with dLight because of variable injection conditions/expression between animals, and this would add an additional requirement of precise timing to make sure a larger/smaller response was not due to recording at a slightly different time point. Although the short duration is noted as a caveat, the implications of this are not.

We appreciate the Reviewer's concern over this important point. To further demonstrate how robust the dLight responses to DETQ we have performed side-by-side comparisons of multiple trials within animal, across animals, as well as across time. As can be seen in our new Figure S11, the dLight responses appear overall very robust and reproducible when compared within animal over multiple injections and over multiple weeks' time. As it can be expected from fiber photometry traces, there is a certain degree of variability in the responses across animals. This is most likely due to differences in total sensor expression levels, optic fiber placement, and other surgery related differences among the different animals. It should be noted that all photometry experiments involving the use of neurotransmitter indicators (namely dLight-family or GRAB-family indicators) suffer from the same issue, as the responses of these indicators are not intrinsically ratiometric (i.e. inter-individual differences cannot be easily normalized). Nevertheless, our results show highly consistent responses to DETQ injection when repeated multiple times within the same animal and over a period of weeks. This indicates that variable

injection conditions as well within-animal differences in expression over a few weeks' time are not importantly affecting the response to DETQ.

We clearly acknowledge the short duration of the DETQ effect as a caveat of this approach in the Discussion of the manuscript, as reported below:

“A limitation of our approach is the short duration of the stable effect that DETQ produces in vivo in mice (i.e. 31 minutes), which is determined by the pharmacokinetic properties of DETQ and i.p. administration route used. Because of this, [...]”

There were also no experiments conducted to confirm that the monitored, amplified signal is DA. dLight has been characterized, but this needs to be done for the response with DETQ.

Good point. We conducted 3 additional experiments to confirm that the dLight responses in the presence of DETQ are indeed due to DA and not to other non-specific signals.

We started from in vitro experiments, by characterizing the molecular specificity of dLight1.3b's response in the presence of DETQ in vitro. The results can be seen in Figure S3e. We applied several non-specific neurotransmitters (Glutamate, GABA, Acetylcholine, Serotonin, Histamine) and showed that the indicator specifically responded only to dopamine, in the presence of DETQ.

We also performed in vivo chemogenetic and optogenetic experiments where we acutely inhibited VTA dopamine neurons after the induction of a step-like response to DETQ in dLight. Our findings, now shown in the new main Figure 6, indicate that transiently inhibiting DA neurons causes a pronounced and dose-dependent, although partial, blunting of the step-like response to DETQ. These results directly and causally involve dopaminergic neuronal activity in the generation of the step-like response to DETQ observed in vivo. The fact that there is a residual response even after chemo/optogenetic inhibition of DA neurons, could indicate that either the silencing is incomplete, or that there are multiple mechanisms underlying the tonic DA levels that are detected with our approach. Indeed much effort is being devoted in the dopamine field to understand the mechanisms of tonic dopamine generation, and several studies implicated activity-independent spontaneous vesicle release as a mechanism that contributes to extracellular dopamine levels (see for example: PMID: 31486769, PMID: 28115487).

We report these findings in the results section and added 1 new figure and 1 new supplementary figure for this (Figure 6 and Figure S3e).

We now expanded the Discussion section with a description of these results, as follows:

“We demonstrate with either chemogenetic (hM4Di) or optogenetic (eNpHR) inhibition of VTA DA neurons that the DETQ-induced step-like increase in dLight1.3b signal in the NAc can be substantially reduced but not completely eliminated with transient inhibition of DA neuron firing, raising the intriguing possibility that both DA neuron activity-dependent and -independent factors contribute to establishing the tonic DA levels detected with our approach in the NAc. In the future researchers could use the approach to mechanistically test the cellular and/or molecular factors involved in determining the amounts of extracellular tonic DA across brain areas, for example by probing the role of spontaneous, action potential-independent DA release mechanisms, with high spatiotemporal precision. Additionally, using this approach future studies could establish the relative changes in both tonic and phasic DA release during different motivational, physiological or disease states, or upon pharmacological stimulation with psychoactive drugs, and correlate them with downstream molecular signals (for example by using cyclic-AMP or kinase activity reporters)”.

Other concerns are outlined below.

1) *The rationale for increasing DA sensitivity needs to be strengthened, given previous evidence that stimulated increases in [DA] are 10-30 μ M (Patriarchi et al 2018).*

We agree with the Reviewer that the rationale for using dLight1.3b potentiation needs to be better clarified. In essence, every DA indicator will exhibit optimal performance within a limited subset of brain areas, due to its pre-determined and fixed affinity and dynamic range. In other words no single indicator can be considered ideal for all applications and all brain areas (one-fits-all). The striatum, which is where the 10-30 μ M [DA] range has been previously estimated by us based on dLight recordings, is a region particularly rich in dopaminergic innervation. Outside of the striatum the situation drastically changes and our approach for boosting the indicator's affinity becomes very useful, if not critical (we showed the example of photometry recordings in the prefrontal cortex in our Figure 8, where with DETQ potentiation we can detect behaviorally-evoked DA release on a single-trial basis, which would not otherwise be possible). We have expanded on the introduction of the manuscript to make this point clear with the following statement:

“However, due to their pre-determined affinity and sensitivity range, existing DA sensors fail to capture well both tonic and phasic extracellular DA levels in a single recording session. Furthermore, each indicator performs optimally within a limited range of extracellular [DA] that is dictated by the indicator's fixed affinity and sensitivity. Thus, an approach that would enable capturing the complexity of endogenous multi-modal DA release and at the same time flexibly enhance the indicator's detection sensitivity would be ideal to complement current DA sensor-based approaches used to investigate DA physiology.”

2) *In this light, the stimulated increases in [DA] reported here (Fig. 3), should already be saturating given the concentration-response curves in Fig. 1b, but yet an increase is dF/F is seen with 200 nM DETQ. These results need to be explained, and the present and previous findings reconciled in the text.*

We would like to point out that the previously published data mentioned by the Reviewer (Patriarchi et al 2018) had been collected using a different indicator (dLight1.2) which has markedly higher affinity and smaller maximal fluorescence response, when compared with dLight1.3b, which we instead used in the current study. In addition, electrical stimulation in the previously published study was provided by a monopolar stimulating electrode which generates larger and more widespread activation of dopamine release sites across the slice, compared to relatively localized activation from the theta glass stimulating electrode (tip diameter: 1.5-2 μ m) used in the present study. As a result, we were able to use a significantly lower stimulus intensity (10 μ A) compared to typical stimuli often used that are orders of magnitude larger (e.g. Salinas et al., 2023, Nat Commun, PMID: 37739964 – using 100-800 μ A). Due to both sensor and stimulation paradigm differences, electrical stimulation data shown in our Figure 3 cannot be directly compared to those reported in Figure 2 of the original dLight article (Patriarchi et al 2018). These two findings are not in disagreement, but rather reflect the fact that different sensors can be deployed for explicitly different purposes. In the case of the dopamine-rich striatum, imaging large electrically stimulated DA responses can be accurately detected using a higher affinity indicator such as dLight1.2. In the current work, we evoked significantly smaller DA responses with the goal of characterizing the ability of DETQ to enhance detection by the lower affinity sensor, dLight1.3b.

In addition, we would like to explain that titration data performed in vitro on HEK293 cells (such as those from (Patriarchi et al 2018) and those in Figure 1 of this work) should only serve as a reference point when making comparisons under similar in vitro conditions. Affinities measured in vitro, will not necessarily match the affinity that the same sensor will have once it is expressed in brain slice, due to the different conditions of the experiments (tissue complexity, different temperature, etc.).

3) 185-187, 218. *Decay time has many contributing factors. Most commonly, a change in decay time is used to suggest a change in the activity of the DA transporter (DAT) or the response time of a probe. As written, the implication is that the longer decay time here reflects the probe's ability to detect falling levels of DA more sensitively, so that DA can be detected for a longer period. If this is the hypothesis, this should be stated explicitly, and evidence presented to show that DETQ does not alter DAT activity or dLight1.3b response time. Indeed, the result shown in Fig. 3a using iontophoresis is very like what one sees when uptake is inhibited (active uptake often prevents detection of exogenously applied DA in the striatum).*

We fully agree with the Reviewer on this important point. To conclusively demonstrate that DETQ does not affect the uptake activity of the endogenous dopamine transporter (DAT), we performed new experiments measuring dopamine release and reuptake with fast-scan voltammetry. The results demonstrate that application of DETQ had no effect on the DA reuptake, while cocaine, which we used as a positive control, did. This indicates that DETQ likely had little to no effect on DAT. We report the findings in the results section and added a new supplementary figure for this (Figure S5).

4) 226-228. *The species selectivity of DETQ for human vs. mouse/rat D1 receptors needs to be clearly explained at the beginning. Without that, the lack of effect on behavior is surprising, as it implies a lack of role for DA in the behaviors examined, or at least the lack of a modulable role. Additionally, the authors do not demonstrate this selectivity for need to demonstrate that this is the case in their hands. They report that DETQ alone does not increase D1-MSN excitability in mouse striatal slices, which provided a critical control. However, more convincing would have been to assess whether DETQ altered the effect of DA on these cells. This should be tested.*

We appreciate the Reviewer's request for increased clarity. We now explain the species selectivity of DETQ on multiple occasions throughout the manuscript, as shown below:

At the end of the introduction (line 84), we wrote:

"Here we demonstrate that a PAM selective for the human dopamine type-1 receptor (hmDRD1) [...]"

At the beginning of the results section (lines 107-116), we wrote:

"[...] because DETQ [...] has a known binding site, and displays potent allosteric activity only on the human DRD1 (K_b-human = 11.4 nM [DETQ]) while it is >30-fold less potent on rodent DRD1 (K_b-mouse = 312 nM [DETQ]), we reasoned that this particular D1-PAM would fit the ideal drug profile to be used for selective chemogenetic potentiation of a human receptor-based DA sensor (Figure 1a)."

We also further reinstated the target selectivity of DETQ in the Discussion with the following paragraph:

"We demonstrated that the species selectivity of DETQ renders it pharmacologically inert on mouse physiology, by characterizing its effects on signaling in vitro using a panel of mouse receptors, by characterizing its effect on endogenous DAT activity in mouse brain slices, and by monitoring locomotion and anxiety-like behavior upon in vivo administration in mice. This evidence adds to an extensive literature in which the lack of effects of this drug on mouse neurochemistry, physiology and behavior had been thoroughly characterized."

We also performed additional in vitro experiments confirming the hmDRD1-selectivity of DETQ over a panel of mouse GPCRs. We tested the effect of DETQ as either allosteric modulator or agonist on these receptors using a Nanoluciferase-complementation assay of mini-G

protein recruitment. Our findings demonstrate that, DETQ only acted as a PAM on the hmDRD1, and had no effect on the other receptors tested, now shown in our new Figure S2. We added a description of these results in the manuscript text, which reads as follows:

“Previous work established the target selectivity of DETQ for hmDRD1 against a series of over 40 human protein targets^{24,25}. To further confirm its selectivity, we tested the effect of DETQ either in allosteric modulator mode or in agonist mode on mini-G protein recruitment for a subset of mouse receptors using a Nanoluciferase (NanoBit)-based complementation assay^{26,27}. Addition of DETQ (100 nM) during DA titrations performed on hmDRD1 and msDRD1, led to a strong and significant potentiation of mini-Gs recruitment only in the case of the hmDRD1, as expected (Figure S2a). No allosteric effect was detected also by applying a high concentration of DETQ (1 μM) on all other mouse receptors tested: DRD2, DRD5, ADRA1a, β2AR and 5HT6R (Figure S2b,c). Furthermore, no direct agonist effect was detected when applying DETQ (1 μM) alone on any of the receptors tested (Figure S2d). Thus, our results support previous findings indicating that DETQ acts selectively as PAM on a hmDRD1 target.”

Furthermore, we performed additional ex vivo patch-clamp experiments demonstrating that DETQ does not alter the DA-sensitivity of mouse D1-MSNs in acute striatal slices. Our results are shown in the new Figure S4g-i. We have updated the results section of the manuscript accordingly.

Lastly, we conducted in vivo experiments where we monitored GCaMP activity of D1-MSNs in the Nucleus Accumbens in response to an aversive stimulus (footshock). While the D1-MSN activity peak could be reduced by administration of a DRD1 antagonist (SCH-23390), indicating the DA-dependence, administration of DETQ did not lead to any alterations in the D1-MSN response, indicating that DETQ does not alter the DA-sensitivity of endogenous mouse DRD1 and the physiology of D1-MSNs in vivo in mice. The results have been added as Figure S8.

5) 259-261. Using a control (AlloLite-ctr) that is relatively insensitive to DA (Fig. 1D) and insensitive to the amplification seen with DETQ to conclude that DETQ does not promote DA release seems circular in that AlloLite-ctr could not detect enhanced baseline release if it happened.

We appreciate the Reviewer's insightful comment. Indeed, both dLight1.3b and AlloLite-ctr are indicators with a relatively low DA affinity, for example when compared to other indicators of the dLight family (their EC50s are approximately 1.6-2 μM). However, and this is critical for the adoption of AlloLite-ctr as a control sensor in the current manuscript, they are almost identical to each other (both in terms of EC50 and dynamic range). Their relatively lower affinity (when compared to other existing indicators) is the price to pay for an exceptionally large dynamic range (in fact, the largest in the dLight family of indicators, about 10-fold increase in fluorescence from baseline to saturating DA in vitro).

The Reviewer is correct in that the absence of a step-like response in the photometry signal of AlloLite-ctr upon DETQ administration to the mice, could lead to two different interpretations, which we initially overlooked:

- i) either the effect observed on dLight1.3b is due to DETQ boosting tonic DA levels to a concentration that can be better detected by dLight1.3b in addition to boosting the sensitivity of dLight1.3b (for example by DETQ having direct effects on dopamine transporter activity), or
- ii) DETQ only achieves this in vivo step-like baseline response by boosting dLight1.3b sensitivity

With our new Fast-Scan Voltammetry experiments we have now conclusively proven that DETQ does not interfere with DAT function, and we can thus confidently state that the lack of

an effect of DETQ on AlloLite-ctr is most likely a direct result of the lack of DETQ potentiation at the indicator.

6) Confusingly, AlloLite is used to refer to dLight1.3b + DETQ in some places, whereas in other places dLight1.3b alone is used as the DETQ-sensitive form. dLight1.3b + DETQ is actually more accurate than the new term, which actually requires two components (and is technically the same as the original sensor). The idea is good in principle, but not in practice if the authors are not consistent; for example, in li. 260, they refer to AlloLite, but mean AlloLite-ctr, which makes a big difference.

We agree with the Reviewer and apologize for the confusing use of a newly coined word instead of the combination of DETQ + dLight1.3b. We have now removed the term AlloLite throughout the manuscript, and left the term AlloLite-ctr for describing the control dLight sensor which does not bind to DETQ. We hope this makes the article easier to follow and understand.

7) Given the results in Figs. 3a,d (showing a many-fold difference for the lowest injection currents), how the % increases in baseline were calculated after DETQ administration needs to be described in the main text, as the starting baseline also appears to be set to zero in Fig. 6c,d.

We have added additional information to methods describing two-photon imaging in acute brain slices. To expand and further clarify, peak $\Delta F/F_0$ was calculated in 'spot photometry' measurements of dLight fluorescence by defining baseline fluorescence (F_0) as the 300 ms immediately before dopamine stimulation/application. This value was subtracted from the entire trace (i.e. $F - F_0$) before changes of fluorescence were normalized to baseline (i.e. $(F - F_0)/F_0$). Peak changes of fluorescence were calculated by maximum peak detection from the time period following dopamine stimulation or application by iontophoresis. Baseline fluorescence (F_0) after DETQ application was defined in the same way – as the 300 ms immediately before dopamine stimulation (in the presence of DETQ).

8) The results in the initial sections of the report describing in vitro studies indicate an 8-fold increase in sensitivity to DA; however, the conclusion after the in vivo studies is that there is a 2-fold difference in signal. The reasons for this may come back to a concern raised earlier about the actual levels of DA detected under different conditions. This should be discussed.

We would like to thank the Reviewer for bringing up this important point. There are two main things to keep in mind:

i) in our in vitro studies we reported an 8-fold potentiation effect on the EC₅₀ of the DA indicator, while in our in vivo studies we reported a 2-fold difference in maximal amplitude of the DA signal evoked by behavioral stimuli – these are two different things: one is a measurement of apparent affinity, the other is a measurement of fluorescent response. We don't have a reason to expect a priori the two effect sizes to match. To avoid confusion on this matter, we now added a statement to the discussion, as follows:

“Our in vitro results demonstrate that DETQ can achieve an 8-fold potentiation effect on the EC₅₀ of dLight1.3b, while in our in vivo studies show that systemic administration of DETQ can boost about 2-fold the amplitude of dLight1.3b's fluorescence response evoked by behavioral stimuli.”

ii) the indicator response to its ligand (DA) follows a non-linear relationship. Hence, an 8-fold increase in indicator's EC₅₀ measured in vitro, non-necessarily will result in a similar fold increase when moving onto a slice or in vivo setting. Important factors that will influence the fold amplitude of the potentiation effect in vivo include the actual baseline [DA] that exists in

the extracellular environment in the brain area under investigation at the moment of the recordings, as well as the amount of DA that is released acutely in response to the external stimulus (e.g. footshock, reward, etc), which varies along with brain area, density of dopaminergic projections, internal and/or learning state of the animal among other things. It is worth noting that a 2-fold enhancement of a DA indicator's signal in vivo is to be considered quite an achievement, given the challenging conditions of in vivo recordings. As an example, second-generation GRAB-DA2 indicators were only able to improve signal amplitude during in vivo recordings by about 2-fold over their first-generation version, both in flies and in rodents (see Sun et al. Nat Methods 2020, PMID: 33087905).

9) A general practice for characterization of any sensor is to conduct experiments to test whether it detects what it is supposed to detect. Although dLight has been well-characterized, the enhanced signals with DETQ have not. This is a concern, particularly for in vivo use. Basic studies to test whether signals detected after DETQ is DA might include local infusion of TTX, or TTX in VTA/SNc to show activity dependence and DA neuron dependence. Such tests often also include pharmacological manipulations that alter DA release in a predictable way (e.g., after DA depletion with reserpine or amplification by cocaine). Here, cocaine was used to show that AlloLite-ctr could detect an amplified DA signal, but this same experiment was not done with AlloLite to show a comparable % increase and change in decay time between AlloLite and dLight1.3b.

As described in our response above, we have now characterized the in vitro molecular specificity of dLight1.3b in the presence of DETQ. Our results show that dLight1.3b retains its molecular specificity in the presence of the D1PAM. We conducted a number of further in vivo experiments to verify that the step-like response of dLight to DETQ administration is indeed DA-dependent. We have discussed the in vivo chemogenetic and optogenetic DA neuron-inhibition experiments that we conducted as well as the results in our response above. Following the Reviewer's suggestion, we have also performed experiments in which we administered cocaine to the animals closely following the administration of DETQ (10 minutes later). The new results, summarized in Figure S9, clearly show that the dLight's response to DETQ is sensitive to cocaine. Particularly, the fluorescence baseline greatly increases and the decay time is prolonged in response to cocaine while the peak amplitude is decreased (as expected: see Figure 2 in PMID: 37739964), even in the presence of DETQ, in a similar manner to what we observed with our AlloLite-ctr recordings.

Minor

45-47. Voltammetric methods have reasonable spatial resolution – but limited to one site at a time. This should be reworded.

We reworded this in the introduction with the following statement:

“The use of experimental techniques such as microdialysis and electrochemistry has provided useful insights on the mechanisms of DA release and reuptake in vivo⁴, but suffers from limitations, as these approaches can only provide single-point measurements and cannot differentiate signals with cellular or subcellular resolution”.

56-57. Cholinergic regulation of axonal DA release has been recognized for at least 20 years, so is not a recent finding.

We corrected this.

124. Should be DA/NE here – implies greater selectivity for NE when written NE/DA.

We corrected this.

167-169. Please specify firing rate “during current injection”, given the well-known lack of firing of MSNs in slices.

We corrected this.

167. Superfusion, not perfusion.

We corrected this.

Generally not necessary to capitalize words for abbreviations, like dorsal striatum (DS) or ventral tegmental area (VTA).

We corrected this.

Terminology should be consistent throughout, for example Allolite and AlloLite are both used, and D1-PAM interleaved with DETQ (if used for variety, use DETQ first in each data section).

We corrected this.

Reviewer #3 (Remarks to the Author):

This manuscript describes the interaction between a D1 PAM and dLight1.3b. The application of DETQ a D1-receptor dependent positive allosteric modulator results in a shift in the concentration required to activate the fluorescence of the dopamine sensor dLight1.3b. The maximal increase in fluorescence is not affected by the PAM. Rather, there is an increase in the affinity of agonists for the sensor. The increased affinity is also responsible for a slowing of the decay following removal of the agonist. The manuscript describes a comprehensive series of experiments that demonstrate the potential utility of using this PAM.

Comments

1. The use of a PAM to modulate the fluorescence increase induced by the activation of the dopamine sensor is an interesting idea that could be useful for experiments aimed at discovering the sites of dopamine release. Beyond that it seems that the use of this method for behavioral experiments will be limited by the action of the PAM on endogenous receptors. As described in the literature, this compound has multiple actions. It is hard to imagine how this approach will be more advantageous than development of a sensor that has higher affinity and/or a larger dynamic range than dLight1.3b.

We would like to thank the Reviewer for the insightful comments and valuable feedback. We would like to start by pointing out that the chemogenetic boosting approach that we introduced in this work provides important applications that cannot be achieved by simply using higher affinity and/or larger dynamic range indicators. In fact, the application of DETQ to the system (whether it is in vitro on cultured DA neurons, on a brain slice or in vivo) allows the experimenter to decide on a timepoint at which it is possible to switch the affinity of the indicator being imaged from a low-affinity state onto a high-affinity state. This approach allowed us for example to monitor and discern as ‘step-like’ increases in fluorescence, changes to tonic [DA] occurring associated with different metabolic states of the animal (new Figure S12), as well as the presence of different tonic DA levels across brain areas (i.e. PFC, dorsal and ventral striatum, please see Figure 7). These types of experiments are currently

not possible using classical approaches for dopamine imaging. Furthermore, we would like to present our argumentations below supporting the safe use of DETQ (at the dose we recommend) in behavioral experiments, without fearing off-target effects. We also would like to point the Reviewer to our answer to point #1 of Reviewer #4, which provides similar context. Based on extensive previous work which we cite in the manuscript, and the new experimental evidence that we obtained over the course of this revision on the characterization of DETQ's effect in vitro and in vivo, we firmly believe that this drug does not affect the mouse physiology at the dose (10 mg/kg) we chose to use and recommend for use to scientists that will want to adopt the approach.

The selectivity of DETQ for the human dopamine D1 receptor (DRD1) is underscored by a multitude of evidence that originates from its discovery and continues through to detailed molecular and pharmacological studies. Initially identified through a meticulous high-throughput screening process from a pool of over half a million compounds, DETQ stood out as a singular hit for D1 positive allosteric modulator (D1PAM) activity, which was subsequently optimized into the compounds DETQ and LY3154207/mevidalen (see Hao et al., 2019, PMID: 31532644). This process alone highlights the specificity of DETQ's action.

Further affirming its targeted selectivity, extensive in vitro testing against over 40 human targets, including a range of GPCRs, demonstrated that DETQ exhibited high target specificity for DRD1 (Svensson et al., 2017, PMID: 27811173; Hao et al., 2019). This was evident as the only other notable activity was a modest inhibition at the 5-HT2B receptor at concentrations far exceeding those required for DRD1 modulation. Also, when tested in potentiator mode against 14 other human targets (including the more closely related D5, D2, Beta-1, Beta-2 and 5-HT6) at 1 μ M, no activity was found (Hao et al., 2019). Additionally, species selectivity studies have been particularly telling, revealing a 30-fold greater potentiation of the human D1 receptor over that of rodents, with human receptors responding with maximum potentiation, unlike their rodent counterparts.

In vivo experiments provided concrete evidence, with humanized D1 knock-in mice showing clear behavioral responses to DETQ, in contrast to wild-type mice, where no response was observed. This distinction was further characterized by the blockade of DETQ's effects by a selective D1 antagonist in these D1 humanized models, indicating a direct and specific interaction of the DETQ with the human DRD1 receptor (Svensson et al 2019, PMID: 31378255; Meltzer et al., 2019, PMID: 30521930; Rajagopal et al., 2024, PMID: 38048913). We also know from previous molecular pharmacology research, such as site-directed mutagenesis, that DETQ uniquely interacts with a particular aminoacid residue of the human D1 receptor that is not shared by rodents (Luderman et al., 2018, PMID: 30068735). In our work, we used this information to engineer a control indicator, AlloLite-ctr, which abolished the DETQ effect on the DA indicator, serving as a decisive control experiment (Figure 5). It showed that the observed fluorescence increase was indeed a result of DETQ's action directly on the DA sensor rather than off-target molecules.

As part of this revision, we performed additional experiments that show how DETQ does not affect the endogenous dopamine transporter, based on new fast-scan cyclic voltammetry data in brain slices (new Figure S5), or a panel of mouse GPCRs that share a close resemblance to DRD1 (new Figures S2), emphasizing that DETQ does not indiscriminately engage with similar structures.

Collectively, we believe this is compelling evidence confirming that DETQ is a highly selective agent for the human DRD1 receptor, mitigating fears of off-target interactions and supporting its use for the chemogenetic boosting of dLight1.3b.

2. The experiments that were used to suggest that DETQ had no direct action on the D1-receptor or dLight1.3b in vitro are convincing. Experiments aimed at the determination of DETQ on endogenous receptors are not convincing. Experiments with the use of a single high concentration of the D1 agonist, SKF38393, on the firing of medium spiny neurons were flawed. Knowing that DETQ did not increase the maximum change in fluorescence it makes one wonder if it would have a similar action on endogenous receptors. Does it change the

concentration response to the D1 agonist? Without concentration response curves to dopamine and/or SKF38393 in the absence and presence of DETQ the interpretation of the results is limited. The same could be said for the experiments with the use of cocaine. Does the sensitivity (dose response) to cocaine change in the presence of DETQ?

To address the effects of DETQ on mouse D1 receptors, electrophysiology experiments were repeated to examine the effects of DA or DA + DETQ on D1-receptor mediated potentiation of medium spiny neuron firing. We found that DETQ had no effect on DA-dependent increases in firing (Figure S4). Within the timeframe of this revision, it was not possible to repeat a full dose response curve, since due to receptor desensitization different slices would be needed for testing each DA concentration, which adds substantially to the amount of effort and animals involved. Instead, we chose a concentration of DA that is known to produce a roughly half-maximal activation of endogenous receptors in brain slices (10 μ M). This concentration was chosen based upon past literature from slices examining D1R signaling (Wei et al., 2017 *J.Physiol*, PMID: 27927785). In addition, our collaborating lab that performed the new electrophysiology experiments for this revision (Prof. Christopher Ford) has extensive experience examining D2R signaling and has found that as opposed to D1Rs, which are thought to have an even lower sensitivity for DA, the EC50 within the dorsal striatum for D2Rs is \sim 10 μ M (Marcott et al., 2018 *Neuron*, PMID 29656874; Gong et al., 2021 *Neuron*, PMID 34506723). This is similar to what has been found by others (Burke & Alvarez, 2022 *Cell Reports*, PMID 35545050).

Lastly, we conducted in vivo experiments where we monitored GCaMP activity of D1-MSNs in the Nucleus Accumbens in response to an aversive stimulus (footshock). While the D1-MSN activity peak could be reduced by administration of a DRD1 antagonist (SCH-23390), indicating the DA-dependence, administration of DETQ did not lead to any alterations in the D1-MSN response, indicating that DETQ does not alter the DA-sensitivity of endogenous mouse DRD1 and the physiology of D1-MSNs in vivo in mice. The results have been added as Figure S8.

3. That GPCRs are complicated molecules even after the manipulations that result in the construction of a transmitter sensor is nicely demonstrated by this work. This approach could be valuable for studies aimed at the detection of sites of dopamine release in experiments with brain slices and potentially in some experiments in vivo, however the interpretation of behavioral experiments will be very limited without extensive controls because of the interaction of the PAM with endogenous receptors.

Please see our response to point #1 above as well as our response to Reviewer #4 – point #1 below.

Reviewer #4 (Remarks to the Author):

Marie and his/her colleagues have developed AlloLite, a sensitive system for detecting dopamine levels that can dynamically respond to low concentrations of DA. As described in their manuscript, AlloLite demonstrates significant benefits, including a broad range of sensitivity to dopamine levels and the ability to distinguish between tonic and phasic dopamine release signals. However, there are still specific questions regarding its practical use and potential applications that I would like the authors to address.

- 1. Previous research has indicated that DETQ acts as a selective positive allosteric modulator (PAM) for the human D1 dopamine receptor (DRD1), exhibiting limited potency in rodents (see reference 17, for instance). The authors have extensively shown that DETQ does not modify mice's behavior in their experiments. However, these findings do not conclusively prove that DETQ's action is restricted to the D1 receptor. There remains the possibility that DETQ might also influence non-dopamine receptors, such as dopamine transporters or other proteins, potentially affecting synaptic dopamine levels. This, in turn, could interfere with the fluorescence signals detected by the AlloLite system. The manuscript would benefit from addressing how the study controls for or mitigates these potential sources of error.*

We would like to thank the Reviewer for the careful consideration of this important point. We provide below a thorough summary of previous and new evidence (obtained with new experiments during this revision) that we believe greatly mitigates these concerns.

Regarding the in vitro target selectivity of the D1PAM for the human DRD1 versus other potential targets (receptor proteins etc):

The D1PAM, DETQ, is part of a group of tetrahydroquinolin (THIQ) chemical scaffold that was discovered via high throughput screening over 500,000 molecules for D1PAM activity. The screening resulted in one single hit (a low potent thienopiperidine) that was then optimized for potency and drug-like activity resulting in DETQ and LY3154207/mevidalen (see Hao et al., 2019, PMID: 31532644). The high selectivity of this chemical scaffold was further established through broad screening in vitro against a series of over 40 human targets including other GPCR's (Svensson et al., 2017, PMID: 27811173; Hao et al., 2019). At a concentration of 10 μ M there was only one target (5-HT2B) where modest activity was observed (48% inhibition of ligand binding) at 10 μ M of DETQ. For comparison, the in vitro potency for DETQ as a D1PAM at the hD1 receptor is 11 nM. Also, when tested in potentiator mode against 14 other human targets (including the more closely related D5, D2, Beta-1, Beta-2 and 5-HT6) at 10 μ M, no activity was found (Hao et al., 2019). In species selectivity studies (Hao et al., 2019; Wang et al., 2018, PMID: 30111649), high selectivity for D1 potentiation (potentiation of dopamine induced activation of cAMP) was found for the human vs the rodent D1 receptors (30x selectivity for human vs mouse D1 in potentiator mode. Also for both DETQ and LY315207, maximal potentiation of dopamine (75-85%) was seen in the human clones, whereas only a partial (45-53% of max potentiation) was seen in the rodent (rat or mouse). This clear species difference is specific for this group of D1PAMs and was also observed in vivo.

Regarding the in vivo species selectivity for D1PAM:

Early work with initial D1PAM leads consistently failed to show activity in vivo in the rodent (Svensson et al., 2017; Hao et al., 2019). This triggered an in depth molecular pharmacology work including site directed mutagenesis and chimera work that identified a rodent-human difference in D1PAM located specifically in the 2nd intracellular loop (Luderman et al., 2018, PMID: 30068735). This finding triggered the development of the humanized D1 knock-in mouse as an in vivo model (Svensson et al., 2017). We have consistently observed lack of activity in normal, wild type (C57BL/6N) mice, when tested in parallel with hD1 mice. At 10 and 30 mg/kg, per os (PO) of DETQ there was no behavioral activity in wild type mice whereas a pronounced behavioral activation was seen in hD1 mice (Svensson et al 2017). Similarly, in wild type mice DETQ (10 mg/kg, PO) did not change brain levels of the histamine metabolites (tele-methylhistamine and tele-methylindolacetic acid (Bruns et al., 2018, PMID: 29102759). In contrast, in hD1 mice, the same dose of DETQ showed significant elevations of these neurochemical endpoints suggesting increased histamine release (Bruns et al., 2018). The behavioral effect of DETQ in hD1 were also blocked by the selective D1 antagonist SCH23390 (Svensson et al 2019, PMID: 31378255; Meltzer et al., 2019, PMID: 30521930; Rajagopal et al., 2024, PMID: 38048913). In further support, Rajagopal et al., (2024, suppl data), recently tested DETQ on behavioral (cognitive and social) endpoints in aged (19 months old) and young (2.5 months old) hD1 and wild type (C57BL/6N) mice. Aged mice (wild-type and hD1)

showed a clear deficit in both Novel Object Recognition and Social Interaction when compared to young mice. DETQ, dosed PO (3-10 mg/kg) fully reversed these deficits in hD1 mice while in wild type aged mice DETQ failed to reverse these deficits, indicating lack of effect of DETQ on cognitive measurements in wild type mice.

As part of this revision we conducted a number of experiments to assess the potential off-target effects of DETQ on a number of molecules, importantly the endogenous dopamine transporter (DAT), as well as on other mouse GPCRs. Using fast-scan cyclic voltammetry in mouse brain slices, we found that DETQ does not affect endogenous DAT activity. Furthermore, we demonstrated through mini-G protein recruitment assays that DETQ did not have any pharmacological effect at a number of mouse GPCRs, including receptors that share sequence and structural homology with DRD1 (i.e. DRD2, DRD5, A1AR, B2AR). Our findings have been added to the manuscript and are summarized in the new Figures S2, S3, S4, and S5.

Additionally, we would like to point out that our control indicator, which we called AlloLite-ctr, was engineered with two point mutations in the DETQ binding pocket, that abolish the effect of the D1PAM on the DA indicator. We used this tool for important control in vivo experiments. The results (Figure 5h-j) show that albeit all other conditions remaining the same, the simple lack of DETQ effect on the DA indicator (by engineered mutations on the target protein) was sufficient to prevent the previously observed increase in indicator's fluorescence baseline upon systemic injection of DETQ. This control experiment, in our opinion, represents the best possible evidence to demonstrate that the step-like increase in indicator baseline reflects the indicator potentiation by DETQ and not a side-effect of DETQ engagement of off-target molecules.

Taken together, the prior literature along with our findings strongly support high selectivity of DETQ for the human DRD1 target and argue against the suggestion that DETQ effects are mediated through interaction with other, unknown, targets.

2. Is there a risk of fluorescence quenching in the AlloLite sensor due to prolonged laser exposure? It would be beneficial for the manuscript to discuss how the decrease in fluorescence intensity, which may occur during the detection process, is accounted for and evaluated in terms of its impact on the experimental outcomes.

In fiber photometry experiments, addressing fluorescence quenching or photobleaching of the indicator due to continuous illumination is an important consideration. To assess the stability of our signals across multiple days, we conducted long-duration (90 min) photometry recordings in the open-field (OF) arena, with an additional 20 min baseline dopamine (DA) signal recording preceding intraperitoneal (i.p.) injection of either vehicle or DETQ. Figure S7 illustrates raw and analyzed data, presenting example traces from both dLight1.3b-expressing and AlloLite-ctr-expressing mice during the OF test. As depicted by the black lines in figures S10a and S10e, representing individual polynomial fits to each channel (465 and 405), the raw signals exhibit notable stability throughout the recording period and remain consistent across experimental days. In our study, we implemented a standard approach to account for small variations in fluorescence intensity due to prolonged excitation. The photometry data were fitted with a 3rd degree polynomial function using MATLAB's polyfit function. Subsequently, the data were normalized and bleaching corrected by dividing each signal by its corresponding fit. Finally, $\Delta F/F_0$ estimation was calculated using the bleaching-corrected data as the difference between 405 nm and 465 nm-excited signals (Figures S10b and S10f). This processing method, which we describe in detail in the methods section of the article, ensures that changes in fluorescence are attributed to experimental conditions rather than technical factors. The stability of raw signals, particularly across experimental days, underscores the effectiveness of our approach in mitigating potential quenching effects on the

measured fluorescent signals following repetitive recordings. The use of individual polynomial fits for each channel further reinforces the reliability and reproducibility of our results.

REVIEWERS' COMMENTS

Reviewer #2 (Remarks to the Author):

The authors have done extensive work to address the extensive but reasonable concerns raised by the original reviewers. Changes in this revision include points of clarification, rewriting and adding sections of the text, and conducting a number of new experiments. Overall, the concerns I had raised have been addressed well. There are still caveats about the method, but these are now better discussed. Overall, the ability to monitor tonic and phasic dopamine signals should be a boon to those who study dopaminergic transmission.

Reviewer #3 (Remarks to the Author):

This manuscript describes the interaction between a D1 PAM and dLight1.3b. The application of DETQ a D1-receptor dependent positive allosteric modulator results in a shift in the concentration required to activate the fluorescence of the dopamine sensor dLight1.3b. The maximal increase in fluorescence is not affected by the PAM. Rather, there is an increase in the affinity of agonists for the sensor. The increased affinity is also responsible for a slowing of the decay following removal of the agonist. The manuscript describes a comprehensive series of experiments that demonstrate the potential utility of using this PAM.

Comments

1. The use of a PAM to modulate the fluorescence increase induced by the activation of the dopamine sensor is an interesting idea that could be useful for experiments aimed at discovering the sites of dopamine release. My confusion with the endogenous D1 receptors in the mouse model is now cleared up with the clear presentation of the fact that the PAM works on the human D1 receptor but not the rodent receptor. It points out that the approach used in this study may be very limited given that species variation in GPCR interaction with PAMs are a necessary step in the use of these constructs.

2. That GPCRs are complicated molecules even after the manipulations that result in the construction of a transmitter sensor is nicely demonstrated by this work. However, I remain unconvinced that this approach will be more generally advantageous than the development of sensors with differing affinities and larger dynamic range.

Reviewer #4 (Remarks to the Author):

The authors have addressed my concerns. I recommend to publish this manuscript.

We would like to thank all Reviewers for their appreciation of the large-scale efforts that we put in the revision of this manuscript. We provide our answers to the remaining Reviewer comments below in blue:

Reviewer #2 (Remarks to the Author):

The authors have done extensive work to address the extensive but reasonable concerns raised by the original reviewers. Changes in this revision include points of clarification, rewriting and adding sections of the text, and conducting a number of new experiments. Overall, the concerns I had raised have been addressed well. There are still caveats about the method, but these are now better discussed. Overall, the ability to monitor tonic and phasic dopamine signals should be a boon to those who study dopaminergic transmission.

We wish to thank the Reviewer for the kind appreciation of the extensive efforts we put into the revision of our manuscript.

Reviewer #3 (Remarks to the Author):

This manuscript describes the interaction between a D1 PAM and dLight1.3b. The application of DETQ a D1-receptor dependent positive allosteric modulator results in a shift in the concentration required to activate the fluorescence of the dopamine sensor dLight1.3b. The maximal increase in fluorescence is not affected by the PAM. Rather, there is an increase in the affinity of agonists for the sensor. The increased affinity is also responsible for a slowing of the decay following removal of the agonist. The manuscript describes a comprehensive series of experiments that demonstrate the potential utility of using this PAM.

Comments

1. The use of a PAM to modulate the fluorescence increase induced by the activation of the dopamine sensor is an interesting idea that could be useful for experiments aimed at discovering the sites of dopamine release. My confusion with the endogenous D1 receptors in the mouse model is now cleared up with the clear presentation of the fact that the PAM works on the human D1 receptor but not the rodent receptor. It points out that the approach used in this study may be very limited given that species variation in GPCR interaction with PAMs are a necessary step in the use of these constructs.

We thank the Reviewer for the careful consideration of our work and for appreciating the usefulness of our approach and the clarifications we provided on its use and limitations in the last rebuttal letter and in the discussion section of the manuscript. Indeed, as we specifically addressed in the manuscript, the use of a PAM to selectively boost the dopamine indicator works well in our case, but is likely not an easily generalizable strategy to other GPCR-sensors, due to the requirement for a drug that is only active on human receptors and not on endogenous receptors of the animal model under investigation. This point is discussed at the beginning of the manuscript's discussion section.

2. That GPCRs are complicated molecules even after the manipulations that result in the construction of a transmitter sensor is nicely demonstrated by this work. However, I remain unconvinced that this approach will be more generally advantageous than the development of sensors with differing affinities and larger dynamic range.

We respect the Reviewer's opinion, but we have a different opinion on this. We believe that our approach is unique in the sense that it allows researchers to readout extracellular tonic DA levels as a step-like response upon DETQ administration, and at the same it provides the high resolution details over phasic DA release that is typical of genetically-encoded dopamine sensors. The ability to monitor tonic and phasic DA simultaneously, as well as longitudinally and across mice, disease states, metabolic states, etc. will open up the opportunity to investigate how are the two modalities of DA release differentially regulated and/or affected by extrinsic and intrinsic factors. We've demonstrated in proof of concept experiments how our approach could be used for detecting food deprivation-induced alterations in tonic DA release (Supplementary Figure 12), as well as for comparing tonic DA levels across brain areas (Figure 7). Future work could make use of the PAM to elucidate changes in tonic DA in specific mouse models of neurodegenerative and psychiatric disorders, as well as in different motivational states. We remain convinced that these applications are simply not possible using high affinity or low affinity indicators alone. In fact, over 10 different types of DA sensors spanning a wide range of affinities and dynamic ranges have been reported (for a review see Labouesse and Patriarchi, *Neuropsychopharmacology*, 2021) since the initial papers describing this technology came out (Patriarchi et al, *Science* 2018; Sun et al, *Cell*, 2018). Yet the available sensors have not been used in applications that can report both tonic and phasic DA simultaneously in the way that we demonstrated in this study. We hope that the Reviewer will agree that ultimately time will tell how transformative this new approach is. We are eager to disseminate this knowledge and see how it will benefit the DA field.

Reviewer #4 (Remarks to the Author):

The authors have addressed my concerns. I recommend to publish this manuscript.

We wish to thank the Reviewer for the kind appreciation of our work and for recommending its publication.